# Mechanism and regulation of cargo entry into the Commander endosomal recycling pathway

Rebeka Butkovič [1] ✉, Alexander P. Walker[1,5], Michael D. Healy [2,5], Kerrie E. McNally [1,4], Meihan Liu [2], Tineke Veenendaal[3], Kohji Kato[1], Nalan Liv [3], Judith Klumperman [3], Brett M. Collins [2] ✉ & Peter J. Cullen [1] ✉

Commander is a multiprotein complex that orchestrates endosomal recycling of integral cargo proteins and is essential for normal development. While the structure of this complex has recently been described, how cargo proteins are selected for Commander-mediated recycling remains unclear. Here we identify the mechanism through which the unstructured carboxy-terminal tail of the cargo adaptor sorting nexin-17 (SNX17) directly binds to the Retriever sub-complex of Commander. SNX17 adopts an autoinhibited conformation where its carboxy-terminal tail occupies the cargo binding groove. Competitive cargo binding overcomes this autoinhibition, promoting SNX17 endosomal residency and the release of the tail for Retriever association. Furthermore, our study establishes the central importance of SNX17-Retriever association in the handover of integrin and lipoprotein receptor cargoes into pre-existing endosomal retrieval sub-domains. In describing the principal mechanism of cargo entry into the Commander recycling pathway we provide key insight into the function and regulation of this evolutionary conserved sorting pathway.

The intracellular eukaryotic endosomal network sorts and transports thousands of integral membrane proteins and their associated proteins and lipids[1–3]. Central to network function are multiprotein complexes that coordinate sequence-dependent recognition of integral proteins with the biogenesis of vesicular and tubular transport carriers[4–6]. Retriever is an essential cargo sorting complex and is a stable heterotrimer of VPS26C, VPS35L and VPS29, that together with the dodecameric CCDC22, CCDC93, COMMD (CCC) complex and DENND10, forms the 16-subunit Commander super-assembly[7–12]. Defects in Commander assembly and function are associated with metabolic disorders including hypercholesterolemia[13–16], viral infection[17,18], and lead to Ritscher-Schinzel syndrome, a multi-system developmental disorder characterized by abnormal craniofacial features, cerebellar hypoplasia, and stunted cardiovascular development[19–21]. In the Commander trafficking pathway sequence-dependent integral protein recognition is principally mediated by the cargo adaptor sorting nexin-17 (SNX17)[9], the FERM domain of which binds to a ØxNxx[Y/F] sorting motif presented in the cytoplasmic facing domains of integral proteins (where Ø is a hydrophobic residue and x is any residue)[22–25]. Over 100 integral proteins require SNX17 and Retriever for their endosomal sorting through the Commander axis including members of the integrin and lipoprotein receptor families[9]. Fundamental to our understanding of the Commander pathway and the dissection of its functional role in health and disease is a central unanswered question: how is SNX17 coupled to

[1]School of Biochemistry, Biomedical Sciences Building, University of Bristol, Bristol, UK. [2]Centre for Cell Biology of Chronic Disease, Institute for Molecular Biosciences, The University of Queensland, SLCA, Australia. [3]Center for Molecular Medicine, University Medical Center Utrecht, Institute of Biomembranes, Utrecht University, Utrecht, The Netherlands. [4]Present address: MRC Laboratory of Molecular Biology, Cambridge, UK. [5]These authors contributed equally: Alexander P. Walker, Michael D. Healy. ✉e-mail: rebeka.butkovic@bristol.ac.uk; b.collins@imb.uq.edu.au; pete.cullen@bristol.ac.uk

Retriever to allow access into the Commander endosomal retrieval and recycling pathway and how is this coupling regulated?

## Results and discussion

### SNX17 associates with Commander via its extended C-terminal domain

Previously we showed that the carboxy-terminal unstructured [465]IGDEDL[470] tail of SNX17, Leu470 being the terminal residue, can bind directly to the PDZ domain of the PDLIM family of proteins[26]. The same sequence is essential for binding to Commander, and we speculated that this was through direct binding to Retriever[9,26]. To test this association, we used biGBac insect cell expression[11] to purify human Retriever and full-length human SNX17 (Supplementary Fig. 1A). When

SNX17 was incubated with nickel affinity resin-bound Retriever we observed specific but non-stoichiometric association (Fig. 1A). Consistent with the requirement of the carboxy-terminal Leu470 residue[9], recombinant SNX17(L470G) bound to Retriever at significantly lower levels (Fig. 1A, Supplementary Fig. 1B). These data establish that SNX17 directly binds to Retriever through a mechanism that involves its unstructured carboxy-terminal tail, a region that is highly conserved across eukaryotic SNX17 (Supplementary Fig. 1C).

Retriever is assembled around a central VPS35L subunit to which VPS26C and VPS29 associate at spatially distant amino- and carboxy-terminal regions of the VPS35L α-solenoid (Fig. 1B)[10–12]. We employed AlphaFold2 modelling[27,28] to predict the association between Retriever and the unstructured human SNX17 tail corresponding to residues

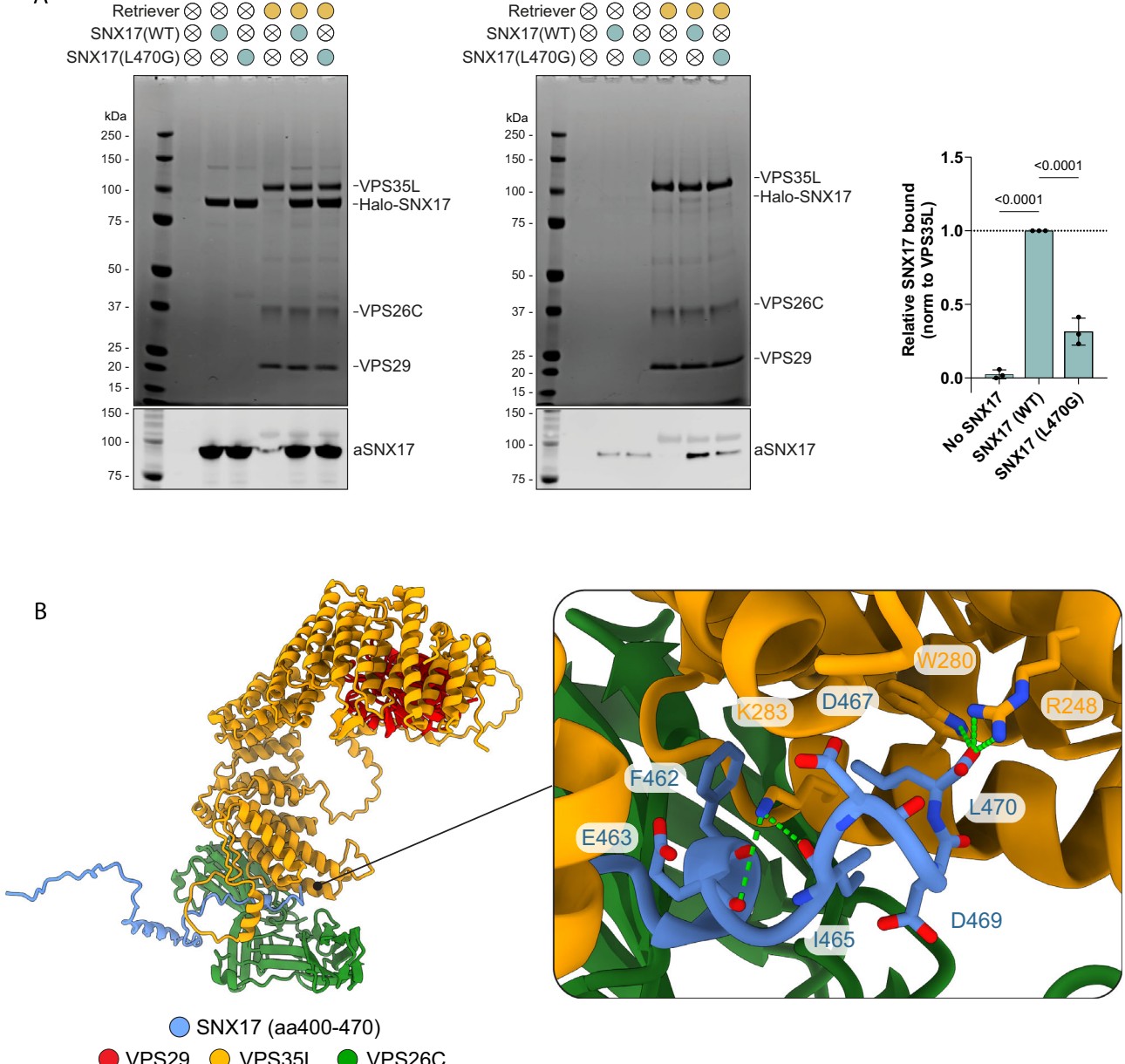

**Fig. 1 | SNX17 directly binds to Retriever. A** Purified His-tagged Retriever was mixed with purified SNX17(WT) or SNX17(L470G) and incubated with anti-His-tag TALON® Superflow beads. Input mixtures (left) and protein bound to the beads after washing (middle) were analysed by SDS-PAGE followed by Coomassie staining and western blotting. SNX17 bound to the beads was quantified and normalised to the level of VPS35L (right). $n = 3$, 1-way ANOVA with Dunnett's multiple comparison test, data presented as mean values and error bars represent s.d. **B** AlphaFold2 predictions show a high confidence interaction between the unstructured carboxy-terminal region of SNX17 and Retriever (Fig S5A).

Gly400-to-Leu470. This predicted a high confidence model where the [465]IGDEDL[470] motif of SNX17 bound to a pocket in VPS35L defined by Arg248, Trp280, and Lys283, that resided close to the VPS35L:VPS26C interface: this pocket is highly conserved across eukaryotic Retriever (Fig. 1B). The interface between SNX17 and Retriever extended to include upstream residues [459]NFAF[462] in the SNX17 tail engaging primarily with VPS26C. One striking feature of the predicted complex is that the extreme carboxy-terminal Leu470 residue of SNX17 makes extensive contact with VPS35L through hydrophobic interaction with Trp280 and via an electrostatic interaction of the carboxy-terminal carboxyl group with Arg248 (Fig. 1B). This agrees with the critical importance of Leu470 for interaction with Retriever (Fig. 1A) and the larger Commander complex in cells[9,26].

Consistent with SNX17 binding to Retriever being a feature of the VPS35L:VPS26C interface, a VPS35L mutant that specifically disrupted binding to VPS26C, VPS35L(R293E)[11], failed to associate with SNX17 in quantitative cell-based immunoprecipitations where mCherry-SNX17 was co-expressed alongside VPS35L-GFP (Fig. 2A). In contrast a mutant that selectively disrupted VPS29 binding to VPS35L, VPS35L(L35D)[11], had no effect on SNX17 association (Fig. 2A). Based on the predicted structure of the SNX17-Retriever complex we performed mutagenesis of the proposed SNX17 binding pocket. These mutations confirmed the AlphaFold2 model; VPS35L(R248A), -(W280A), and -(K283E) all showed a pronounced loss of SNX17 association (Fig. 2B). Highlighting the selectivity of these mutations, all VPS35L mutants targeting SNX17 binding retained assembly into Retriever, binding to the CCC complex, and assembly into Commander.

To further validate the AlphaFold2 model, we performed targeted mutagenesis of the SNX17 [459]NFAF[462] and [465]IGDEDL[470] sequences. Quantitative GFP-trap experiments in transiently transfected HEK293T cells established that GFP-SNX17(I465A), -(D467A) and -(L470G) mutants led to near-complete loss of Retriever binding (Fig. 2C). Similarly, SNX17(NFAF-AAAA) and the more conservative SNX17(F462A) and -(F462E) mutants also displayed reduced Retriever and CCC complex association (Fig. 2D). In contrast, SNX17(D469A) and, more modestly SNX17(E468A) also impacted on the Retriever association (Fig. 2C), consistent with the predicted structure where these sidechains make no direct contacts with the Retriever complex (Fig. 1B). Collectively these data show that Retriever coupling of SNX17 is mediated by motifs within its unstructured carboxy termini binding to surfaces at the VPS26C:VPS35L interface of Retriever. Modeling the association of Retriever and Snx17 from *Drosophila melanogaster* (Supplementary Fig. 2B) and other species such as zebrafish (not shown) predict essentially identical interactions between SNX17 and the Retriever assembly, supporting the evolutionary conservation of the coupling mechanism.

Although the molecular details are all very different, at a general level this coupling mechanism is reminiscent of how the related Retromer complex engages its own cargo adaptors sorting nexin-3 (SNX3) and sorting nexin-27 (SNX27): an unstructured region of SNX3 binding to the VPS26A/B:VPS35 interface[29–31] and the PDZ domain of SNX27 binding to VPS26A/B[32,33] (Supplementary Fig. 2A). While the relative orientation of Retriever to the endosomal membrane has yet to be resolved, we speculate that the similarity in SNX3-Retromer binding, and the ability of SNX17 to associate with PI(3)P, indicates that the SNX17 binding VPS26C-VPS35L interface likely lies in close proximity to the membrane surface (Supplementary Fig. 2A).

## The SNX17-Commander interaction is required for cargo recycling to the plasma membrane

To test the functional importance of direct SNX17-Retriever coupling we performed rescue experiments in a VPS35L CRISPR/Cas9 knock-out RPE1 cell line. Re-expressed wild-type VPS35L-GFP localised to the endosomal network as expected, as did the

VPS35L(R248A) mutant which associates normally to Retriever and the CCC complex but is defective in binding to SNX17 (Fig. 3A). Previous studies have shown that loss of VPS35L expression leads to Commander dysfunction including a reduction in the association of endogenous COMMD1, a marker of the CCC complex and Commander assembly to Retromer labelled endosomes[11]. This phenotype was rescued by re-expression of either VPS35L-GFP or VPS35L(R248A)-GFP although there was a slight but significant trend towards reduced COMMD1 recruitment for the R248A mutant (Fig. 3A). At the functional level, VPS35L KO leads to a reduction in the steady-state cell surface enrichment of α5β1-integrin and the mis-sorting of the internalized integrin into LAMP1-positive late endosomes/lysosomes[9], but does not significantly perturb whole-cell levels of SNX17 (Supplementary Fig. 2C). In imaging experiments these phenotypes were fully rescued by re-expression of wild-type VPS35L-GFP but not by the SNX17-binding defective VPS35L(R248A)-GFP mutant (Fig. 3B). Biochemical quantification of whole-cell and cell surface α5β1-integrin confirmed its enhanced lysosomal-mediated degradation (Fig. 3C, Supplementary Fig. 2D) and reduced steady-state plasma membrane level caused by SNX17 binding deficiency, consistent with reduced recycling[9,24] and extended the significance of the coupling mechanism to another functionally important SNX17 cargo the lipoprotein receptor LRP1[22] (Fig. 3C and Supplementary Fig. 2D). Together, these data reveal the core mechanism of SNX17 coupling to Retriever and the essential importance of coupling for endosomal retrieval and recycling of SNX17 selected cargoes through the Commander pathway.

## A SNX17 autoinhibitory sequence regulates cargo and Retriever interactions

We next explored how the coupling between SNX17 and Retriever could be regulated. The FERM domain of SNX17 comprises three submodules, F1, F2 and F3[25]. Alphafold2 modelling suggested that cargo proteins carrying the conserved ØxNxx[Y/F] sorting motif, including β1-integrin and LRP1, bind through β-sheet augmentation in a complementary groove of the F3 module, in agreement with previous X-ray crystallographic structures (*e.g.* P-selectin (PDB:4GXB) (Fig. 4A). In models of apo SNX17 from various species (data not shown), the carboxy-terminal [459]NFAF[462] sequence (human numbering) invariably formed an intramolecular association with the F3 groove, mimicking cargo sequences in what would be an autoinhibitory conformation (Fig. 4B). The carboxy-terminal [465]IGDEDL[470] also adopted an intramolecular conformation that would be mutually exclusive of Retriever binding (Fig. 4B). In our initial modelling of the SNX17 interaction with Retriever we used only the isolated carboxy-terminal disordered region, which yields essentially identical structure predictions across all five models (Fig. 1B, Supplementary Fig. 5A). However, when we expanded the AlphaFold2 modelling to incorporate full-length SNX17 only two of the five predicted structures displayed the same mechanism of SNX17 binding to the VPS26C:VPS35L interface (Supplementary Fig. 3A). The other three predicted structures showed no interaction with Retriever; in these cases, the [459]NFAF[462] sequence of SNX17 adopted the intramolecular association as seen in apo SNX17 predictions (Fig. 4B; and Supplementary Fig. 3A). This led us to speculate that such a conformation may reflect an autoinhibited state that negatively regulates coupling to Retriever and hypothesized that such autoinhibition could be released by competitive binding of ØxNxx[Y/F]-containing cargo.

To test this hypothesis, we quantified the association of LRP1, LRP2, and APP, model ØxNxx[Y/F] cargos, to SNX17 wild-type and mutants targeting the conserved intramolecular [459]NFAF[462] motif (Supplementary Fig. 1C): SNX17(NFAF-AAAA), SNX17(F462A) and -(F462E). In all cases, immuno-isolation of GFP-tagged SNX17 mutants from HEK293T cells revealed a robust enhancement of LRP1, LRP2, and

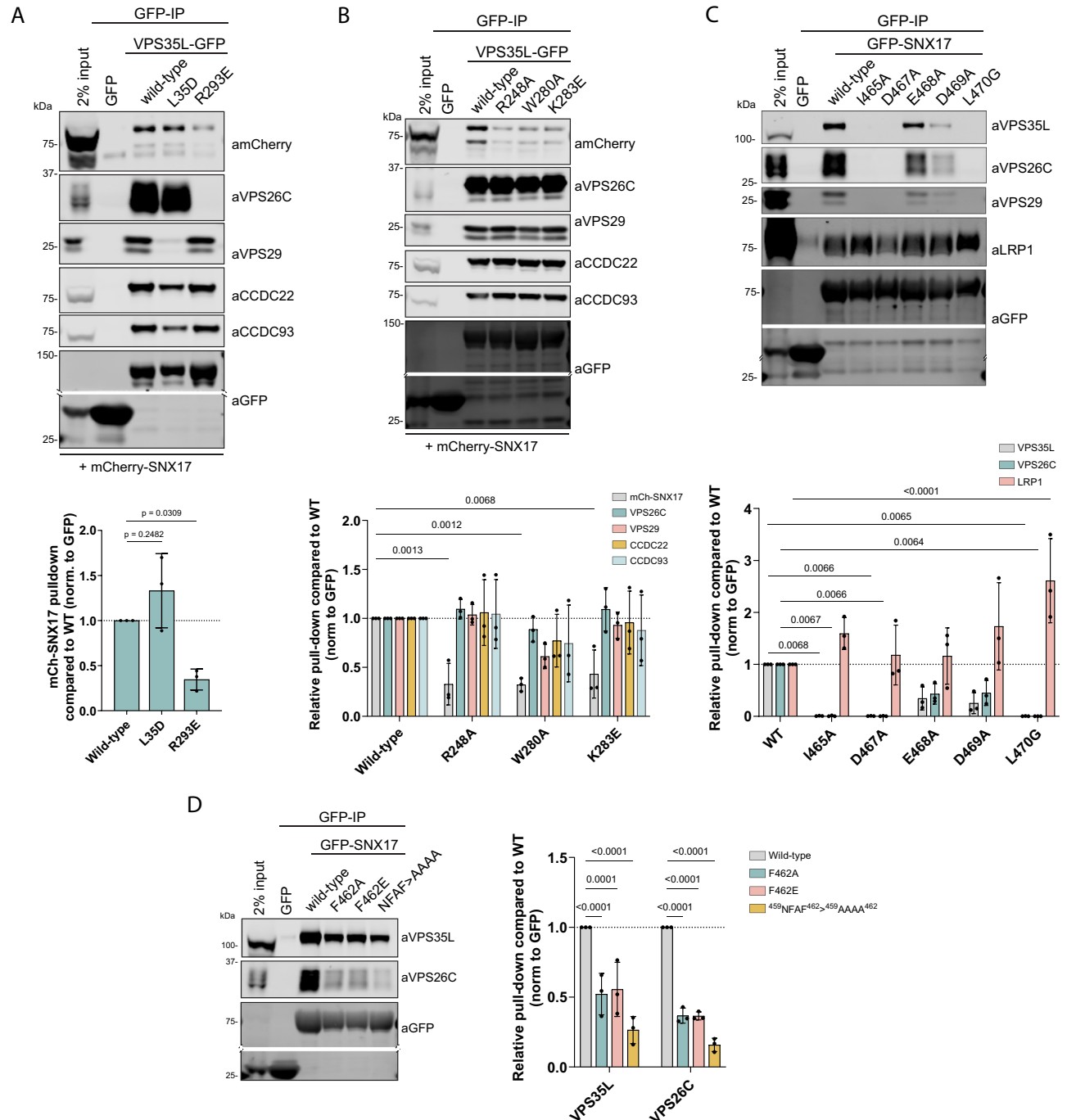

**Fig. 2 | SNX17 binds to the VPS35L-VPS26C interface of Retriever. A** HEK293T cells were transiently co-transfected with mCherry-SNX17 and either GFP, VPS35L-GFP or VPS35L-GFP mutants that perturb VPS35L-VPS26C (VPS35L(R248A)) and VPS35L-VPS29 (VPS35L(L35D)) associations prior to GFP-nanotrap isolation and quantitative western blot analysis of protein band intensities. $n = 3$, 1-way ANOVA with Dunnett's multiple comparison test, data presented as mean values and error bars represent s.d. **B** HEK293T cells were transiently co-transfected with GFP, or VPS35L-GFP or VPS35L-GFP mutants that target SNX17 binding, and mCherry-SNX17. Protein lysates were then used in GFP-nanotrap experiments. Below, the quantitative analysis of protein band intensities is shown. $n = 3$, 2-way ANOVA with Dunnett's multiple comparison test, data presented as mean values and error bars represent s.d., only changes with $p < 0.05$ are shown. **C, D** HEK293T cells were transiently co-transfected with GFP or GFP-SNX17 or GFP-SNX17 mutants in all conserved residues of the terminal $^{465}$IGDEDL$^{470}$ motif (**C**) or conserved $^{459}$NFAF$^{462}$ motif (**D**) that target Retriever binding. Protein lysates were then used in GFP-nanotrap experiments. Below, the quantitative analysis of protein band intensities is shown. $n = 3$, 2-way ANOVA with Dunnett's multiple comparison test, data presented as mean values and error bars represent s.d., only changes with $p < 0.05$ are shown.

APP binding compared to the wild-type protein (Fig. 4C). This suggests that weakening or ablating the intramolecular interaction of the $^{459}$NFAF$^{462}$ sequence allows for increased intermolecular binding of ØxNxx[Y/F]-containing cargo to the FERM domain. To extend this, we designed a set of complementary mutations in the SNX17 FERM domain required for binding to ØxNxx[Y/F]-containing cargo[25]; we reasoned that these would relieve the autoinhibition with the carboxy-terminal $^{459}$NFAF$^{462}$ sequence and thus enhance Retriever association. Indeed, the GFP-tagged SNX17(W321A), -(V380D) and -(M384E) mutants all displayed a modest but significant increase in binding to

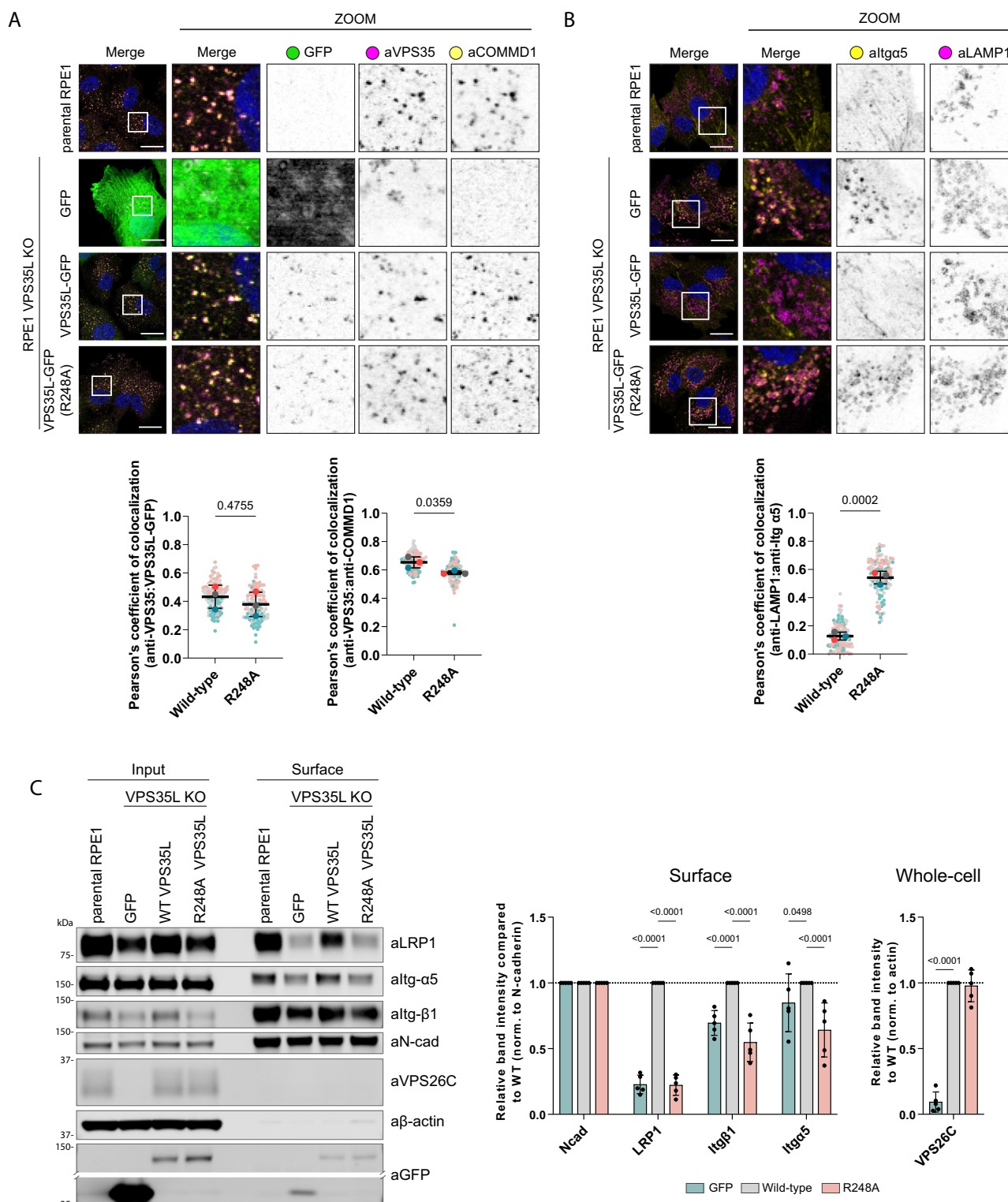

Retriever and the Commander super-assembly (Fig. 4D). These mutants also showed a reduced binding to the LRP1 cargo, consistent with binding being mediated through the same groove.

To further explore cargo binding in the competitive release of the autoinhibition, we modelled the association of full-length SNX17 with the cytosolic tail of LRP1 using AlphaFold2. This showed the expected β-sheet augmentation of the LRP1 $^{4470}$NPTY$^{4473}$ sequence with the FERM domain[25] and a displacement of the intramolecular $^{459}$NFAF$^{462}$ interaction (Fig. 5A). To directly demonstrate the presence of an auto-inhibitory interaction we next preformed isothermal titration

calorimetry (ITC) to quantify the association of LRP1 and APP cargo peptides (containing $^{4470}$NPTY$^{4473}$ and $^{759}$NPTY$^{762}$ ØxNxx[Y/F] motifs respectively) with recombinant full-length SNX17 and a SNX17 deletion mutant lacking the carboxy-terminal tail predicted to form the intra-molecular inhibition (residues 1-390) (SNX17ΔC). Consistent with the autoinhibitory model and the carboxy-terminal tail interfering with cargo binding, both LRP1 and APP peptides displayed an approximate 2-fold lower affinity for binding to full-length SNX17 when compared with SNX17ΔC (Fig. 5B, Supplementary Table 1). In addition, SNX17ΔC displayed very weak affinity binding to a synthetic peptide of the

**Fig. 3 | SNX17-Retriever coupling is essential for Retriever-cargo retrieval to plasma membrane. A** VPS35L KO RPE1 cells were lentivirally transduced with GFP, VPS35L-GFP or VPS35L-GFP(R248A). The stably expressed VPS35L(R248A) localizes to endosomes and can partially rescue COMMD1 localization. Scale bars correspond to 20 μm. Pearson's coefficients were quantified from 3 independent experiments (GFP-VPS35 coloc. wt: *n* = 114 cells, R248A: *n* = 107 cells, VPS35-COMMD1 coloc. wt: *n* = 117 cells, R248A: *n* = 106 cells). Pearson's coefficients for individual cells and means are presented by smaller and larger circles, respectively, colored according to the independent experiment. The means (*n* = 3) were compared using a two-tailed unpaired t-test. Error bars represent the mean, s.d. **B** VPS35L KO RPE1 cells were lentivirally transduced with GFP, VPS35L-GFP or VPS35L-GFP(R248A). The stably expressed VPS35L(R248A) failed to rescue Itgα5 missorting as evidenced by increased co-localisation with lysosomal marker LAMP1.

Scale bars correspond to 20 μm. Pearson's coefficients were quantified from 3 independent experiments (wt: *n* = 125 cells, R248A: *n* = 113 cells). Pearson's coefficients for individual cells and means are presented by smaller and larger circles, respectively, colored according to the independent experiment. The means (*n* = 3) were compared using a two-tailed unpaired t-test. Error bars represent the mean, s.d. **C** Cell surface proteins were biotinylated in stably rescued RPE1 cells and enriched with streptavidin pull-down to analyze surface protein levels. The VPS35L(R248A) mutant failed to rescue cell surface levels of Itgα5, Itgβ1 and LRP1, but stabilised VPS26C. The quantitative analysis of protein band intensities is shown. The band intensities were normalised to the respective cell surface N-cadherin levels. *n* = 5, 2-way ANOVA with Dunnett's multiple comparison test, data presented as mean values and error bars represent s.d., only changes with *p* < 0.05 are shown.

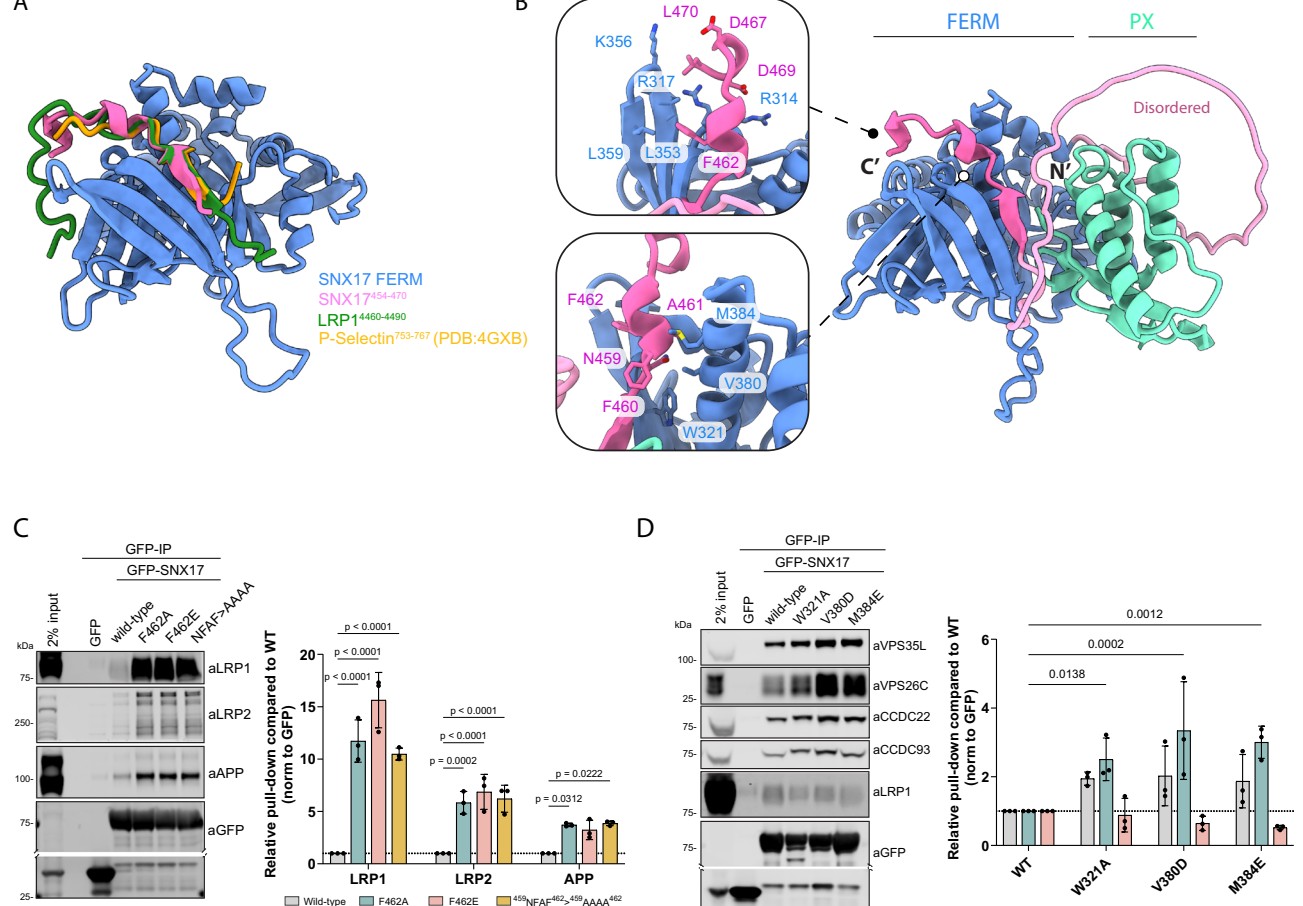

**Fig. 4 | Intramolecular association between SNX17 FERM domain and $^{459}$NFAF$^{462}$ autoinhibits SNX17 binding to cargo and Retriever. A** Overlay of the FERM domain of SNX17 bound to P-selectin (PDB: 4GXB), LRP1 (Supplementary Fig. 5C) and the intramolecular SNX17 peptide (Supplementary Fig. 5B) (as predicted by AlphaFold2) showing a clear overlap in peptide occupancy within the canonical cargo binding pocket. **B** The C-terminus of SNX17 contains an $^{459}$NFAF$^{462}$ motif highlighted in bright pink that is predicted by AlphaFold2 (Supplementary Fig. 5B) to bind into the canonical cargo binding pocket, in addition several residues in the extreme C-terminus ($^{467}$DEDL$^{470}$) are predicted to stabilize this interaction. **C** HEK293T cells were transiently co-transfected with GFP, or GFP-SNX17 or GFP-SNX17 mutants in the $^{459}$NFAF$^{462}$ motif to target its intra-molecular association with

SNX17-FERM domain. Protein lysates were then used in GFP-nanotrap experiments. Below, the quantitative analysis of protein band intensities is shown. *n* = 3, 2-way ANOVA with Dunnett's multiple comparison test, data presented as mean values and error bars represent s.d. **D** HEK293T cells were transiently co-transfected with GFP, or GFP-SNX17 or GFP-SNX17 mutants in the FERM(F3) domain to target its intra-molecular association with the$^{459}$NFAF$^{462}$ motif. Protein lysates were then used in GFP-nanotrap experiments. Below, the quantitative analysis of protein band intensities is shown. *n* = 3, 1-way ANOVA with Dunnett's multiple comparison test, data presented as mean values and error bars represent s.d., only changes with *p* < 0.05 are shown.

carboxy-terminal tail of SNX17, an interaction that was no longer observable when using full-length SNX17 or SNX17ΔC that had been pre-incubated with the LRP1 peptide (Fig. 5C, Supplementary Table 1). Finally, to independently validate the ITC data, we purified recombinant Retriever and full-length SNX17 and reconstituted their

association in the presence of a synthetic peptide corresponding to the LRP1 cytoplasmic tail containing the $^{4470}$NPTY$^{4473}$ ØxNxx[Y/F] motif. SNX17 associated with Retriever coated beads, and this association was enhanced in a dose-dependent manner by inclusion of the LRP1 peptide (Fig. 5D, Supplementary Fig. 3B). In control experiments, a

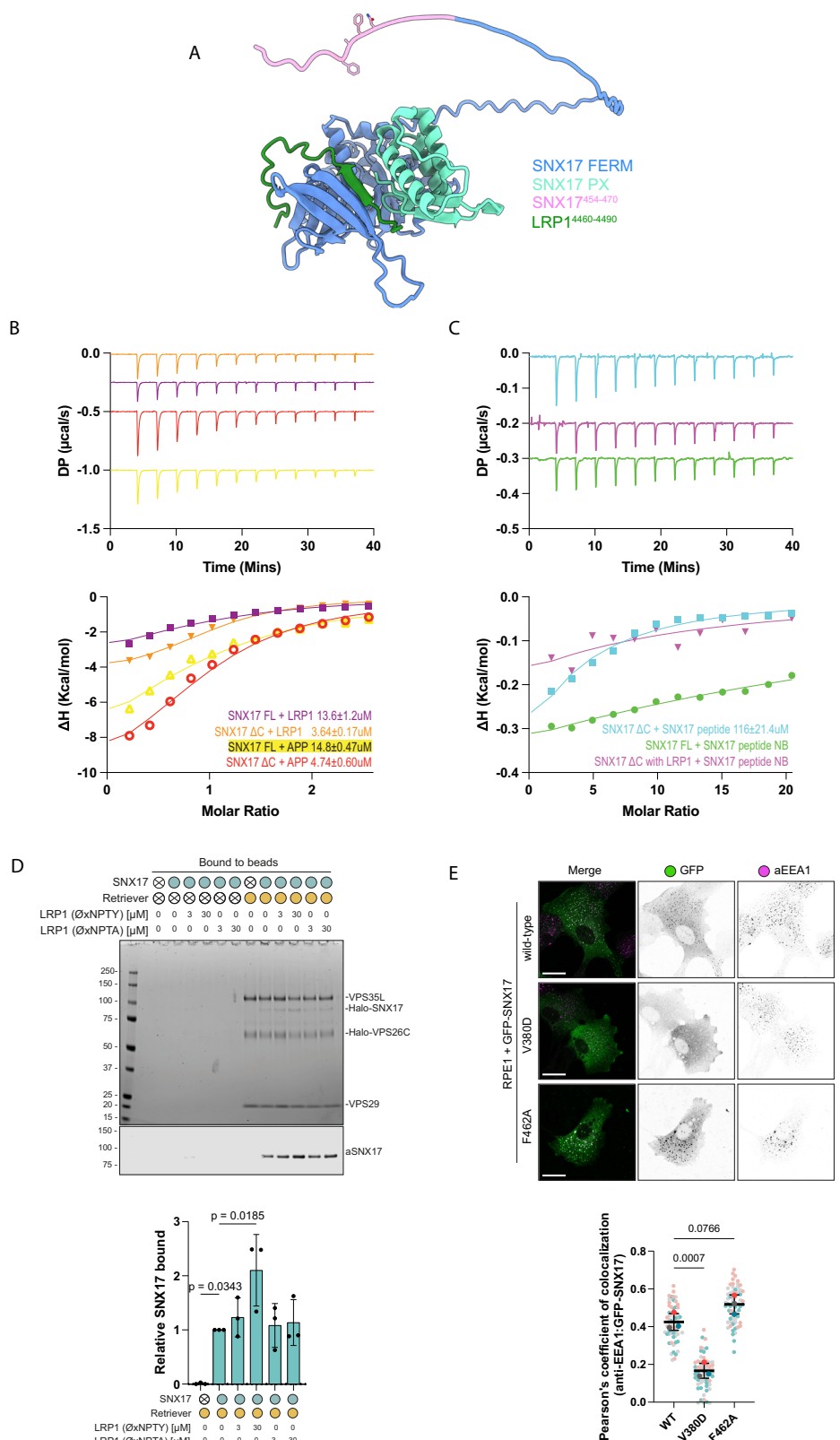

corresponding LRP1 peptide carrying a Y4473A mutation in the ØxNxx[Y/F] motif failed to enhance SNX17 association to Retriever (Fig. 5D, Supplementary Fig. 3B). Collectively, these data support the proposed model whereby an autoinhibited SNX17 conformation is released through a competitive interaction with ØxNxx[Y/F] cargo, which allows for the subsequent association of SNX17 and bound cargos with Retriever and the Commander super-assembly.

## SNX17 is not enriched with Commander at endosomal retrieval sub-domains

A number of studies have shown that specific endosomal membrane sub-domains are enriched for either degradative components such as ESCRT or recycling machinery including Retromer, Retriever, the CCC complex, and WASH-dependent actin patches[9,34–44] (for review[1]). We therefore explored the importance of SNX17 coupling to Retriever and

**Fig. 5 | Cargo binding to SNX17 relieves autoinhibition and promotes association with Retriever. A** Modelling of full-length SNX17 with the cytoplasmic tail of LRP1[4460-4490] predicts a perturbation in the intramolecular interaction defined by LRP1 preferentially binding to the cargo binding pocket releasing the disordered carboxy-tail of SNX17. **B** Isothermal titration calorimetry was performed with the cargo recognition peptide of [4466]LRP1[4479] and [755]APP[768] against either the full-length SNX17 (residues 1-470) or a construct which lacked the C-terminal tail (residues 1-390) (SNX17ΔC). In each case the peptide showed an ~2-fold decrease in binding when the C-terminal tail was absent. **C** Using a synthetic peptide of the C-terminus of [452]SNX17[470] we confirmed that this region can interact with SNX17ΔC, albeit with relatively low affinity, and that this interaction was inhibited when using the full-length construct or the SNX17ΔC construct following pre-incubation with LRP1. $K_d \pm$ SEM was calculated from triplicate data. **D** Purified His-tagged Retriever was mixed with purified SNX17 (WT) and 3 μM–30 μM of LRP1 (NPXY) or LRP1 (NPXA) peptide. The mixtures were incubated with anti-His-tag TALON® Superflow beads, then input mixtures and protein bound to the beads after washing were analyzed by SDS-PAGE followed by Coomassie staining and western blotting. SNX17 bound to beads was quantified and normalised to the level of VPS35L (right). $n = 3$, 1-way ANOVA with Dunnett's multiple comparison test, data presented as mean values and error bars represent s.d., only changes with $p < 0.05$ are shown. **E** RPE1 cells were transiently transfected with GFP-SNX17 wild-type, or SNX17(V380D) or SNX17(F462A) mutants that decrease or enhance cargo binding, respectively. Fixed cells were examined with confocal microscope, and the localisation of the GFP-tagged constructs was compared to the localisation of early endosome marker EEA1. Scale bars correspond to 20 μm. Pearson's coefficients were quantified from 3 independent experiments (wt: $n = 62$ cells, V380D: $n = 60$ cells, F462A $n = 64$ cells). Pearson's coefficients for individual cells and means are presented by smaller and larger circles, respectively, coloured according to the independent experiment. The means ($n = 3$) were compared using a 1-way ANOVA with Dunnett's multiple comparison test. Error bars represent the mean, s.d.

cargo for their endosomal association and sub-domain organization. Like other cargo binding coat complexes, for example AP2[45], the endosomal localization of SNX17 requires avidity-based co-incident sensing of phosphoinositides, specifically phosphatidylinositol 3-monophosphate (PI(3)P) and ØxNxx[Y/F]-containing cargo proteins[25]. Consistent with this, the FERM-domain mutant GFP-SNX17(V380D) that exhibited decreased cargo-binding failed to localize to the endosomal network when transiently transfected into RPE1 cells (Fig. 5E). In contrast, the enhanced cargo-binding GFP-SNX17(F462A) mutant displayed an even more pronounced endosomal association, as evidenced by increased co-localisation with the endosomal marker EEA1 compared with wild-type GFP-SNX17 (Fig. 5E). These results suggest that SNX17 can cycle between the cytoplasm and the endosomal membrane through low affinity sensing of PI(3)P with the density of incoming ØxNxx[Y/F]-containing cargo providing an additional affinity to prolong the endosomal residency of SNX17 and release the autoinhibited conformation thereby facilitating coupling to Retriever and the Commander assembly.

To examine the endosomal organization of SNX17 and its relationship to Retriever association we used a VPS35L KO cell line engineered to stably express control GFP, or re-express VPS35L-GFP or VPS35L(R248A)-GFP – a mutant that retains Retriever assembly but inhibits binding to SNX17 (Fig. 2B) thereby uncoupling SNX17 mediated cargo recognition from the downstream process of cargo recycling (Fig. 3B, C). To evaluate the relative organization of VPS35L-GFP with endogenous SNX17 and other endosomal markers, we analyzed multiple endosomes through confocal immunofluorescence imaging, and plotted the normalized average fluorescence intensity profiles to evaluate the relative distributions of endosomal markers within a single endosome. Endogenous SNX17 displayed a general distribution over the bulk of EEA1-positive endosomal membranes (Fig. 6A). In contrast VPS35L-GFP and VPS35L(R248A)-GFP were enriched on one or more foci of the SNX17-labelled endosomes, and co-localized with endogenous COMMD1 (Fig. 6B) and FAM21 (Supplementary Fig. 4A), markers of the CCC and WASH complexes respectively. Retriever, CCC and WASH complex markers also co-localized with the core Retromer component VPS35 and SNX1, an ESCPE-1 subunit that drives endosomal tubulation during the biogenesis of transport carriers[46-48] (Fig. 6C, and Supplementary Fig. 4B). These foci therefore represent the previously described retrieval sub-domains from where cargo-enriched transport carriers exit the endosome for transport to their destination[9,34,35,39,41,49]. Our data shows that the association of SNX17 (and cargo) with Retriever was not a pre-requisite for sub-domain organization, as VPS35L(R248A)-GFP retained localization to the retrieval sub-domain. We also noticed that VPS35L KO cells displayed a partial perturbation in the sub-domain organization of FAM21, SNX1 and VPS35 (Fig. 6C, and Supplementary Fig. 4A, B). Interestingly, in VPS35L KO cells, these markers occupied multiple, less well-defined foci on the EEA1 and SNX17 endosomal membrane with greater localization overlap with EEA1 or SNX17 (Fig. 6C, and Supplementary Fig. 4A, B).

To independently confirm these observations, we turned to an ultrastructural analysis using immuno-electron microscopy. Fixation and immunogold staining of the VPS35L KO RPE1 cell line engineered to stably re-express VPS35L-GFP revealed enrichment of Retriever on a tubular-vesicular sub-domain extending from the vacuolar endosomal limiting membrane (Fig. 7A). On the same individual endosome, endogenous SNX17 was enriched predominantly on the vacuolar region (Fig. 7A). In the parental wild-type RPE1 cells, immunogold staining for endogenous VPS35 and SNX17 confirmed the enrichment of SNX17 on the vacuolar endosomal region and showed the enrichment of VPS35 (Retromer) to a tubular-vesicular sub-domain that was morphological indistinguishable from the Retriever labelled sub-domain (Fig. 7B i-ii). Together these data confirm the clear spatial separation of SNX17 from the Retriever and Retromer labelled retrieval sub-domain. Finally, in the VPS35L KO RPE1 cell line (not rescued by VPS35L-GFP re-expression) we observed the loss of spatial restriction of VPS35 to the tubular sub-domain, rather it localized alongside SNX17 on the vacuolar region of the endosomal limiting membrane (Fig. 7B iii–iv).

Altogether these data indicate: (i) that VPS35L and the Retriever complex play a role in organizing endosomal retrieval sub-domains; (ii) that cargo sensing by SNX17 can promote its recruitment to EEA1-positive endosomes but is not a pre-requisite for the formation of the retrieval sub-domain; and (iii) whilst present, SNX17 is not enriched in these retrieval sub-domains. Rather it appears that the transient coupling of cargo bound SNX17 to Retriever and Commander may serve to handover cargo into a pre-existing retrieval sub-domain for endosomal exit and recycling to the cell surface (Fig. 8).

In summary, by identifying the molecular details of SNX17 coupling to Retriever we have revealed the evolutionarily conserved mechanism through which hundreds of integral membrane proteins, including integrins and lipoprotein receptors, enter the Commander retrieval and recycling pathway. Importantly, we establish that cargo binding facilitates SNX17 endosomal association, and that cargo occupancy relieves an autoinhibited SNX17 conformation to promote Retriever association and the entry of cargo into a pre-existing retrieval sub-domain for the promotion of Commander mediated cell surface recycling. Overall, our study provides fundamental mechanistic and regulatory insight into the role of the SNX17-dependent Commander retrieval and recycling pathway during essential cellular processes ranging from directed cell migration through to cholesterol homeostasis.

Further experiments will be required to broaden our mechanistic understanding of the dynamics of cargo handover into the retrieval sub-domain and the biogenesis of tubular exit gates, how these events are controlled in response to changes in the cellular state, and how

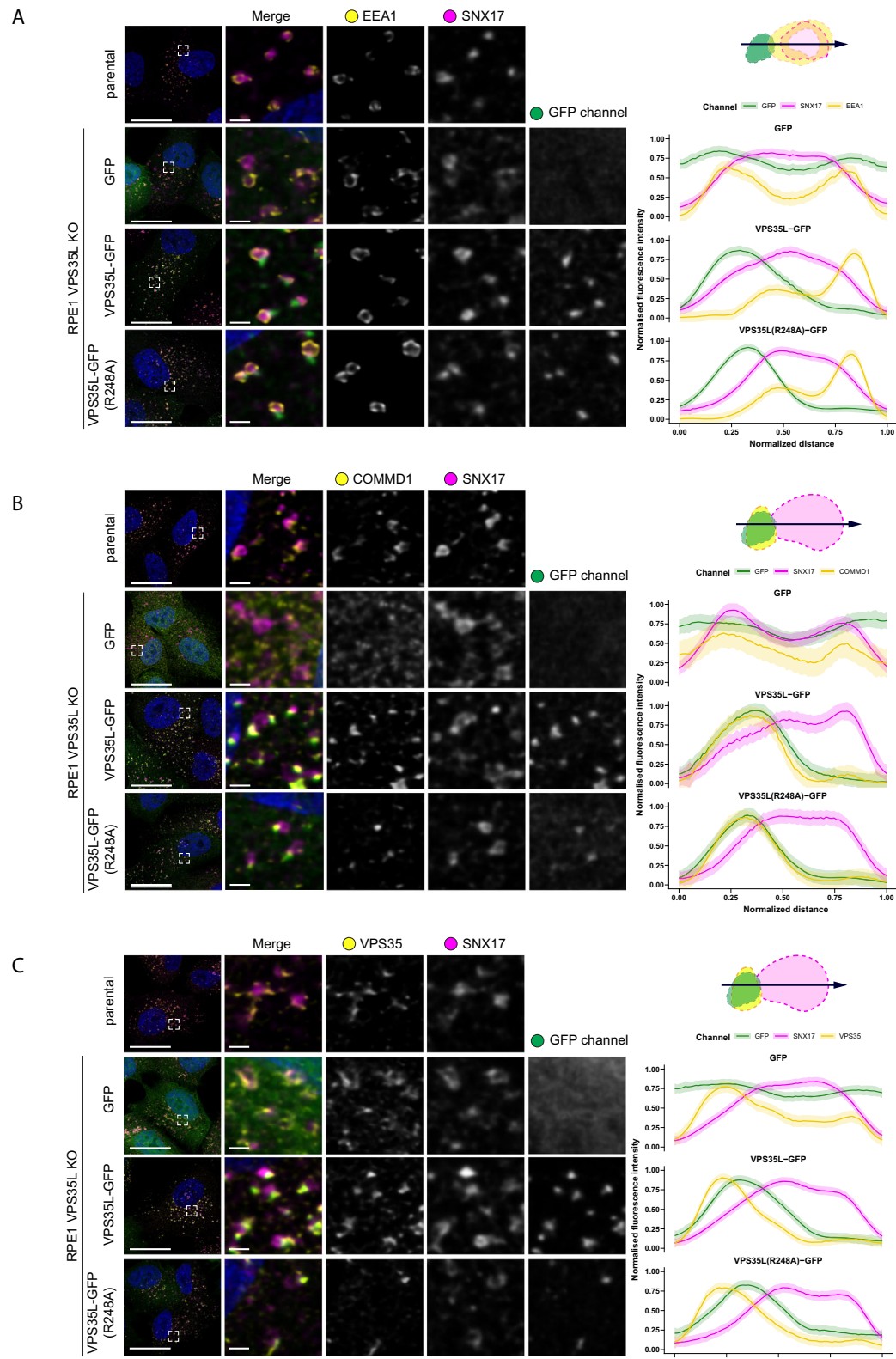

**Fig. 6 | Retriever resides on recycling sub-domain of endosome and colocalises with the markers of the CCC complex. A–C** VPS35L KO RPE1 cells were lentiviral transduced with GFP, VPS35L-GFP or VPS35L-GFP(R248A). The localisation of GFP-tagged proteins was compared to the localisation of endogenous endosomal markers SNX17, EEA1 (**A**), COMMD1 (**B**) or VPS35 (**C**). Representative confocal microscopy images are shown. The relative distributions of endosomal markers were evaluated in ImageJ by generating fluorescence intensity line profiles. Line profiles of 30 endosomes from 3 independent experiments were analysed in Rstudio, where the lengths of line scans and raw fluorescence intensities were normalised and averaged. The average profiles are shown on the right. Loess curve with 95% confidence interval. Scale bars shown for full image or inset correspond to 20 μm and 2 μm, respectively.

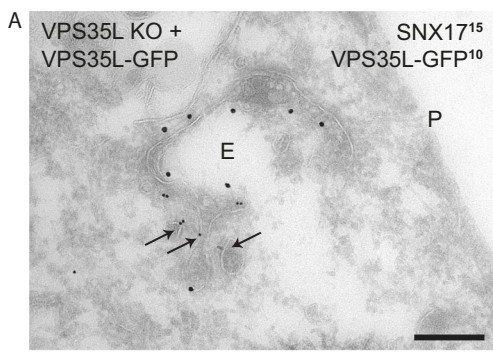

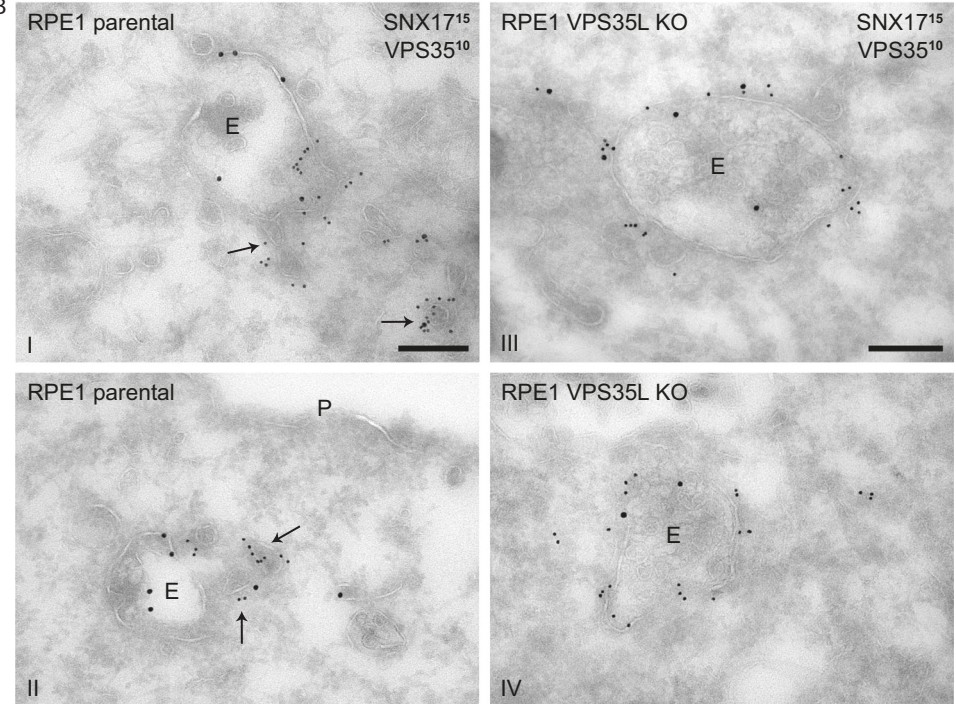

**Fig. 7 | Retriever depletion perturbs Retromer sub-domain organisation.**
**A** Double immunogold labelling of endogenous SNX17 (15-nm gold) and VPS35L tagged with GFP (10-nm gold) in VPS35L KO RPE1 cells expressing wild-type VPS35L-GFP, showing spatial separation of SNX17 and VPS35L over vacuolar (E) and tubular (arrows) endosomal subdomains, respectively. A representative micrograph is shown (2 technical replicates). **B** Double immunogold labelling of endogenous SNX17 (15-nm gold) and VPS35 (10-nm gold) in parental RPE1 (panels **i–ii**) and VPS35L KO RPE1 cells (panels **iii–iv**). In parental cells, VPS35 (Retromer) is localized to endosomal tubules (arrows), whereas in VPS35L KO cells the majority of VPS35 was found on SNX17-decorated endosomal vacuoles. Representative micrographs are shown (2 technical replicates). Scale bars correspond to 200 nm. E = Endosomal vacuole.

Retriever regulates Retromer sub-domain organization and functional cargo retrieval and recycling.

## Methods
### Antibodies and materials
Primary antibodies against the following targets were used: APP (Abcam, ab32136), CCDC22 (Proteintech, 16636), CCDC93 (LSBio, C336997), COMMD1 (Sigma Aldrich, WHO150684M1), EEA1 (BD Biosciences, 610457), FAM21 (Gift from Dan Billadeau), GFP (WB; Roche, 11814460001), GFP (immuno-EM; Rockland, 600-401-215), Itgα5 (Abcam, ab150361), Itgβ1 (BD Biosciences, 610467), LAMP1 (Developmental Studies Hybridoma Bank, H4A3), LRP1 (Abcam, ab92544), LRP2 (Proteintech, 19700-1-AP), mCherry (Antibodies.com, A85306), N-cadherin (Cell Signalling Technology, (13A9) 14215), SNX1 (BD biosciences, 611482), SNX17 (IF/Immuno-EM; Sigma, HPA043867), SNX17 (WB; Proteintech, 10275), VPS26C (Millipore/Sigma Aldrich, ABN87), VPS29 (Santa Cruz, SC-398874), VPS35 (IF; Antibodies.com, A83699),

VPS35 (immuno-EM; Santa Cruz, sc-374372), VPS35L (Abcam, ab97889), β-actin (Sigma Aldrich, A1978).

For secondary detection the following antibodies were used: Goat anti-Mouse IgG (H + L) Cross-Adsorbed Secondary Antibody Alexa Fluor 680 (Invitrogen, A-21057), Goat anti-Rabbit IgG (H + L) Secondary Antibody DyLight 800 (Invitrogen, SA5-35571), Alexa Fluor 568 anti-mouse IgG (Invitrogen, A10037), Alexa Fluor 568 anti-rabbit IgG (Invitrogen, A10042), Alexa Fluor 647 anti-mouse IgG (Invitrogen, A31571), Alexa Fluor 647 anti-rabbit IgG (Invitrogen, A31573), Alexa Fluor 647 anti-goat IgG (Invitrogen, A21447).

Bafilomycin A was purchased from Tocris Bioscience.

### Cell culture
HEK293T and RPE1-hTERT cell lines were maintained in DMEM (Sigma, D5796) with 10% (v/v) fetal bovine serum (Sigma, F7526) in presence of penicillin/streptomycin (Gibco) at 37 °C with 5% $CO_2$.

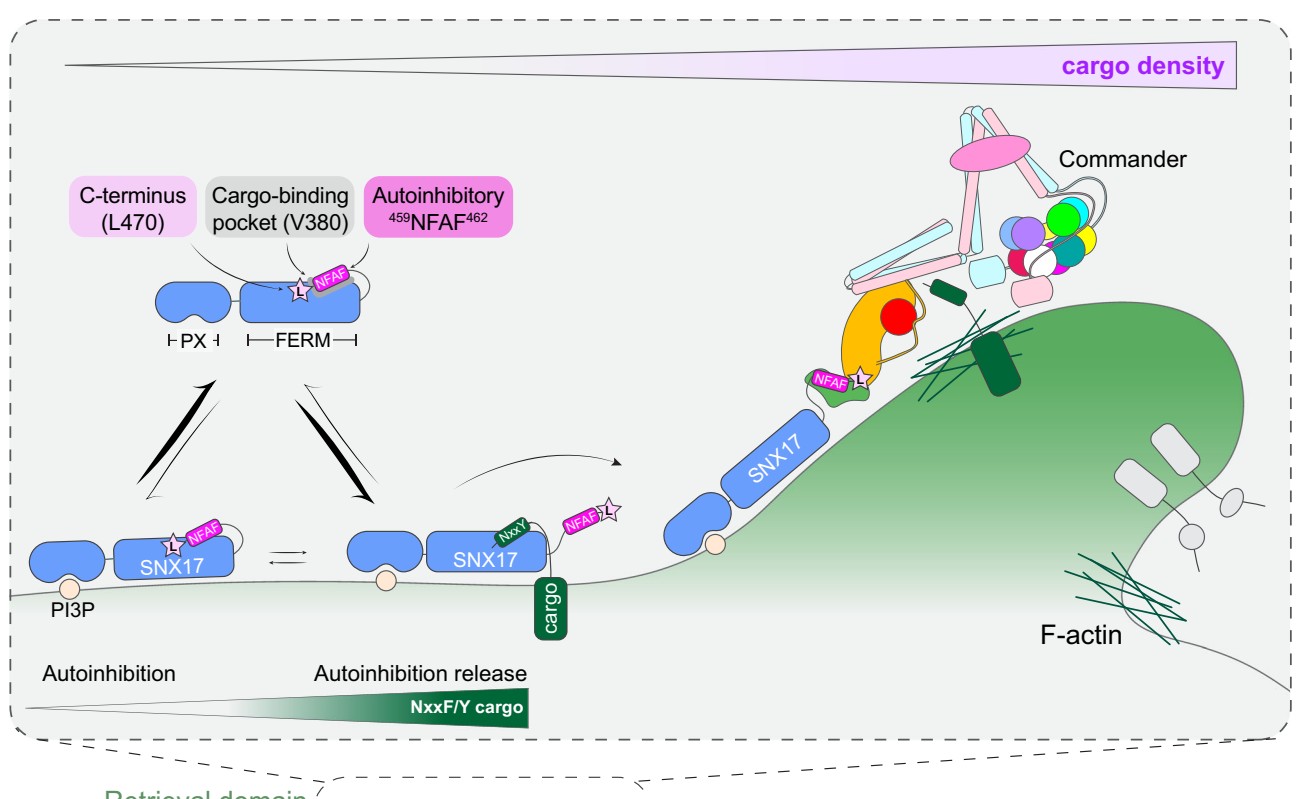

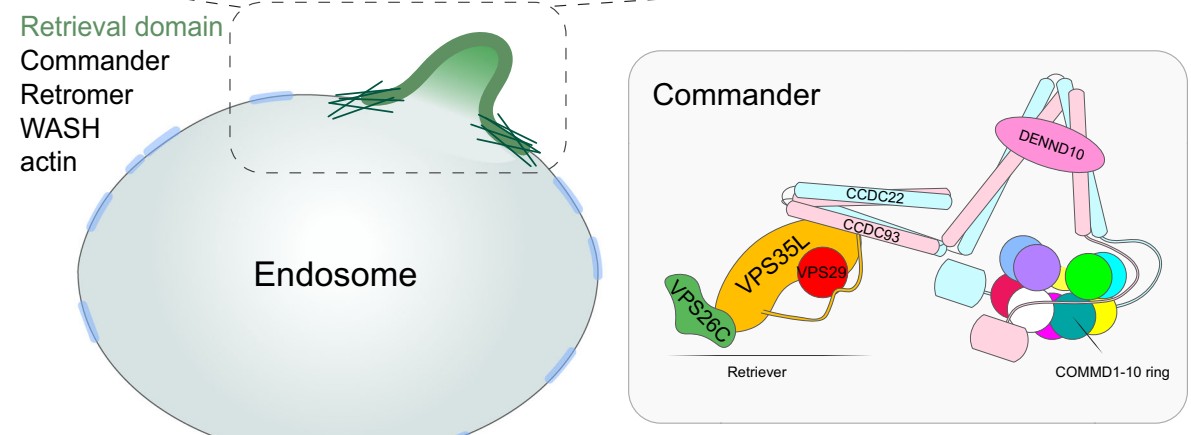

**Fig. 8 | Model of SNX17-Commander association and its regulation through ØxNxx[Y/F] cargo-density sensing and endosomal sub-domain organization.** SNX17 associates with the endosomal membrane enriched for phosphatidylinositol 3-monophosphate (PI3P). We hypothesize that endosomal localization is enhanced through the binding of transmembrane cargoes containing ØxNxx[Y/F] sorting motifs. With increasing cargo density, the auto-inhibitory conformation defined by the SNX17 tail associating with the cargo binding groove in the FERM domain of SNX17 is displaced, enabling the presentation of SNX17 carboxy-tail to the conserved VPS26C:VPS35L interface. The direct binding of cargo-bound SNX17 to Retriever ultimately leads to Commander-mediated recycling of cargo back to the plasma membrane. For simplicity, other endosomal sorting complexes, such as the F-actin polymerizing WASH complex, ESCPE-1 and Retromer are not shown.

## VPS35L CRISPR KO cell line generation

A RPE1-hTERT cell line lacking VPS35L was generated using CRISPR/Cas9 technology. The target sequence of CCTGTTTCTTGTTCGA-GAGCTTC on exon28 (NM_020314.7) was inserted into the px458 plasmid (Addgene plasmid no. 48138) and transfected to cells using PEI. After 48 hrs of transfection, GFP-positive cells were sorted by FACS for single clone isolation. VPS35L-KO was confirmed by WB analysis.

## Site-directed mutagenesis

Primers for site-directed mutagenesis were designed using Agilent QuikChange Primer design tool. QuikChange II Site-Directed Mutagenesis Kit (Agilent, 200523-5) was used for mutagensis of VPS35L-GFP constructs following the manufacturer's protocol. All other site-directed mutagenesis was carried out using Q5 High-Fidelity 2X

Master Mix (NEB, M0492) following the manufacturer's protocol. Nonmutated template DNA was digested after the PCR SDM reactions by 1 h incubation with Dpn1 enzyme at 37 °C. This DNA was used for bacterial transformation into XL10 Gold (Agilent, 200315) chemically competent cells according to manufacturer's instructions. The bacteria were grown on appropriate antibiotic-containing agar plates. Full open-reading frames were sequenced to verify the results of mutagenesis. Primer sequences are listed in Supplementary Data 1.

## Gibson assembly

To subclone VPS35L-GFP into lentiviral pLVX vector, wild-type or mutant VPS35L sequences were amplified using Q5 High-Fidelity 2X Master Mix (NEB, M0492) following the manufacturer's protocol. After amplification, PCR samples were resolved on agarose gel and amplified

fragments extracted from gel using GFX PCR DNA and Gel Band purification kit (GE Healthcare, 28-9034-70). The purified DNA or 1 μg of plasmid DNA were digested with XmaI and XbaI for 1 h at 37 °C in 1x CutSmart buffer and nuclease-free water. To eliminate self-ligation, the plasmid was also treated with 1.5 μl of quick-CIP (NEB, M0525). Following digestion, DNA was again purified as above and a ligation reaction between 1:6 ratio of backbone:vector was carried out with T4 DNA ligase (Invitrogen, 15224017). Ligation mixture was incubated at 10, 20 and 30 °C for 30 s, for a total of 200 cycles.

## Recombinant protein expression and purification

Retriever with a VPS29-His tag and a Strep-VPS26C or Halo-VPS26C tag was expressed in Sf21 insect cells and purified as described previously (11). A gene encoding full-length human SNX17 with an N-terminal Strep-tag and HaloTag® was codon optimised for S. *frugiperda* and purchased from Twist Biosciences (San Francisco, CA). The gene was cloned into pACEBac1 using BamHI/HindIII restriction sites and the resulting pACEBac1-Strep-Halo-SNX17 vector was used to generate V1 and V2 baculoviruses as described previously (11). For protein expression, 4 mL V1 or V2 virus was added to Sf21 cells at a density of $0.5\text{-}1.0\text{x}10^6\text{mL}$ in 400 mL Sf-900 II SFM (Thermo). Cells were harvested 72 h post-infection and cell pellets were stored at −80 °C until use. For SNX17 purification, insect cells were resuspended in cold lysis buffer (50 mM HEPES pH 7.2, 150 mM NaCl, 2 mM β-mercaptoethanol, 0.1% (v/v) Triton X-100, EDTA-free protease inhibitor (Pierce)) and lysed on ice using a 130-Watt Ultrasonic Processor (Cole-Palmer) for 2 mins 30 s using a 10s-on 30s-off cycle. Insoluble material was removed by centrifugation at 20000 rpm in a JA-20 fixed-angle rotor (Beckman Coulter) for 25 mins at 4 °C, then the soluble lysate was incubated with 1 mL equilibrated Strep-Tactin® Sepharose® resin (IBA Lifesciences) rotating for 1 h at 4 °C. After binding, the resin was washed 3x in lysis buffer and protein was eluted with a further 3 washes in elution buffer (50 mM HEPES pH7.2, 150 mM NaCl, 2 mM β-mercaptoethanol, EDTA-free protease inhibitor (Pierce), 2.5 mM desthiobiotin (IBA Lifesciences)). Eluates containing protein were concentrated to <500 μL and size-exclusion was performed at 4 °C using an ÄKTA Pure Protein Purification System (GE healthcare) and Superose® 6 Increase 100/300 GL column (Cytiva), with 0.5 mL fractions collected using an F9-C Fraction Collector (Cytiva). Fractions containing SNX17 were pooled and concentrated.

SNX17 full length (residue 1-470) and SNX17ΔC (residue 1-390) were synthesised with an intramolecular decaHis tag between residue 335 and 346 by GeneUniversal and subcloned into pET28a (XbaI and XhoI). GST-LRP1 was available in a pgex4T-2 vector, GST was produced from expression of a native pGEX6P-1 vector[25]. These plasmids were transformed into *E.coli* BL21 DE3 competent cell (New England Biolabs) and plated on agar plates containing Ampicillin or Kanamycin. Clones from this agar plate were grown overnight in 30 mL of LB broth. 5 mL from these cultures was added to 1 L of LB supplemented with 40 mM $NH_4Cl$, 4 mM $MgCl_2$, 4 mM $NaSO_4$, 2.5% glycerol and 30 mM α-lactose and grown at 25 °C for 24 h. Cells were harvested by centrifugation at $6000 \times g$ for 10 min at 4 °C and the harvested cell pellet was resuspened in 50 mM Tris pH 8.0, 500 mM NaCl, 5 mM imidazole, 2 mM β-mercaptoethanol, 10% glycerol, 50 μg/mL benzamidine and 100 units of DNaseI. Cells were lysed by cell disruption at 35 kPSI and clarified by centrifugation $50,000 \times g$ for 30 mins at 4 °C. Talon or glutathione Sepharose (Clonetech) was used to isolated SNX17 constructs and GST constructs, respectively. His tagged constructs were eluted via 500 mM imidazole and GST-tagged proteins were removes via 50 mM glutathione, other buffer components were as above. These eluted proteins were subsequently passed through a superdex s75 16/60 column attached to an AKTA Pure system (GE healthcare) in 50 mM Tris pH 8.0 and 300 mM NaCl.

## In vitro SNX17-Retriever interaction assays

0.1 mg/mL purified Retriever with a VPS29-His tag was mixed with 0.075 mg/mL purified SNX17 (WT) or SNX17 (L470G) in a total volume of 0.1 mL cold lysis buffer (50 mM HEPES pH 7.2, 150 mM NaCl, 2 mM β-mercaptoethanol, 0.1% (v/v) Triton X-100, EDTA-free protease inhibitor (Pierce)). LRP1 (NPXY) (Biotin-GRMTNGAMNVEIGNPTYKMYEG GEPDDG) and LRP1 (NPXA) (Biotin-GRMTNGAMNVEIGNPTAKMY EGGEPDDG) peptides corresponding to human LRP1 amino acid residues 4458–4483 were purchased from GenScript and, where indicated, added to a final concentration of 3 μM or 30 μM. Protein mixtures were incubated with 20 μL equilibrated TALON® Superflow beads (Cytiva) rotating for 1 h at 4 °C, then beads were washed 3x in cold wash buffer (50 mM HEPES pH 7.2, 150 mM NaCl, 2 mM β-mercaptoethanol, 0.1% (v/v) Triton X-100, 10 mM imidazole, EDTA-free protease inhibitor (Pierce)). Washed beads were resuspended in 4x SDS loading dye + 2.5% β-mercaptoethanol for analysis by SDS PAGE followed by Coomassie staining and western blotting.

In the presence of 20 μL of glutathione Sepharose (Clonetech) 2 nmol SNX17 or SNX17ΔC was mixed with 1 nmol GST-LRP1 for 5, 10, 30 or 60 mins and 1 nmol GST for 60 mins. At each time point the protein resin mixture was centrifuged at $5000 \times g$ for 1 min and the supernatant was removed to stop further interaction. After 60 min glutathione resin was washed 5 times with a buffer containing 20 mM Tris pH 8.0, 300 mM NaCl, 2 mM β-mercaptoethanol, 10% glycerol, 0.5% igepal and EDTA-free protease inhibitor (Pierce). All buffer was removed, and beads were resuspended in 30 μL of buffer (100 NaCl, 20 mM Tris pH 8.0) and 10 μL 4x SDS loading dye before incubation at 100 °C for 5 min. 10 μL of sample was loaded onto precast SDS-PAGE gels (Novex) and after running for 43 min at 165 v were stained using Coomassie blue (Sigma).

## Isothermal titration calorimetry (ITC) methods

The affinities of SNX17 full length (1-470) and SNX17ΔC (1-390) against synthetic peptide of the SNX17 tail (452–470) and cargo recognition sequence of APP (755-768) and LRP1 (4466-4479) was determined using a Microcal PEAQ instrument (Malvern, UK). Experiments were performed in 500 mM NaCl 100 mM Tris pH 8.0 and 5% glycerol. APP and LRP1 peptides at 250 μM were injected into 20 μM SNX17 FL or SNX17ΔC, while 2 mM SNX17 tail peptide was injected into 20 μM SNX17 FL, SNX17ΔC or SNX17ΔC saturated with LRP1 (from a previous ITC run). In all cases 13 × 3.22 μL aliquots were used at a temperature of 10 °C. The dissociation constants ($K_d$), enthalpy of binding (ΔH) and stoichiometries (N) were obtained after fitting the integrated and normalised data to a single site binding model. The apparent binding free energy (ΔG) and entropy (ΔS) were calculated from the relationships $\Delta G = RT\ln(K_d)$ and $\Delta G = \Delta H - T\Delta S$. All experiments were performed at least in triplicate to check for reproducibility of the data.

## Transient transfection

For GFP trap experiments and lentiviral particle production, HEK293T cells were transiently transfected using PEI (polyethyleneimine). 10 ml of Opti-MEM were equally split between 2 sterile tubes. The first tube was used to dilute 15 μg plasmid DNA (or as described in 'Lentiviral particle generation'), and the second to dilute PEI at 3:1 PEI:DNA ratio. The PEI dilution was sterilised through 0.2 μm filter. The total contents of both tubes were then mixed and incubated for 15 min prior to tranfection of cells. The cells were incubated with DNA:PEI mixture in Opti-MEM for 6 h, and after incubation, this was replaced for normal growth media for 24 or 48 h. For imaging experiments, RPE1-hTERT cells were transfected using Lipofectamine LTX Reagent with PLUS Reagent (Invitrogen, A12621) according to manufacturer's instructions. Briefly, 200 μl of Opti-MEM were split equally between 2 sterile tubes. The first tube was used to dilute 0.5 μg plasmid DNA in the presence of 3 μl Plus reagent. The other tube was

used to dilute 3 μl of Lipofecatmine. The total contents of both tubes were then mixed and incubated for 5 min at room temperature. Following the incubation, transfection mixture was added drop wise to cells grown on coverslips in 6-well plates and cells were grown for 24 h before fixation.

### Lentiviral particle and stable cell line generation

To generate lentivirus, HEK293T cells were grown in 15 cm dishes and transfected with 15 μg of PAX2, 5 μg pMD2.G and 20 μg of lentiviral expression vector using PEI transfection. After the 48 h incubation, the growth media containing the lentivirus was harvested and filtered through a 0.45 μm filter. 6-wells of VPS35L KO RPE1-hTERT cells were transduced at 25% confluency with varying volumes of lentiviral media. After 48 h, cells were treated with 15 μg/ml of puromycin to select cells with successfully integrated lentiviral constructs.

### Quantitative GFP-nanotrap

Transiently transfected HEK293T cells, expressing GFP or GFP-tagged proteins were rinsed twice with ice-cold PBS and then lysed in a buffer containing 0.5 % NP-40, 50 mM Tris pH 7.5 in ddH$_2$O (in GFP-SNX17 pull-downs) or PBS (co-transfections). Supernatant was collected after 10 min centrifugation at 15000 rpm. 30 μl of supernatant was removed to serve as whole-cell input and diluted in 1:1 ratio with 4x loading buffer containing 2.5 % β-mercaptoethanol. The remaining supernatant was incubated with 25 μl of GFP-nanotrap (gta-20, Chromotek) beads for 1 h at 4 °C. The beads were then washed twice in a buffer containing 0.25 % NP-40, 50 mM Tris pH7.5 in PBS and once in 50 mM Tris pH 7.5 in PBS (or diluted in ddH$_2$O for GFP-SNX17 pull-downs). Between the washes, the beads were collected at the bottom of the tube by 1 min centrifugation at 2000 rpm. Beads were resuspended in a 2x loading buffer, containing 2.5 % β-mercaptoethanol and all samples were denatured at 95 °C for 10 min.

### Cell surface protein biotinylation

RPE1-hTERT cells were washed generously with ice-cold PBS prior to labelling with cell-impermeable 0.2 mg/mL Sulfo-NHS-SS Biotin (ThermoFisher Scientific, no. 21217) in PBS (pH 7.4). The biotinylation reaction was then quenched by incubating the cells in TBS for 10 min, During the labelling, quenching and washing steps, cells were incubated on ice to prevent endocytosis and unspecific labelling of intracellular proteins. After quenching, cells were lysed in lysis buffer containing 2% triton x-100 with protease inhibitors in PBS. Supernatant was collected after 10 min centrifugation at 15000 rpm, and total protein levels analysed using BCA reaction. Inputs were collected and stored separately. Equal amounts of protein were incubated with Streptavidin beads (GE Healthcare, USA) for 30 min at 4 °C. The samples were then washed once in PBS with 1% triton x-100, twice in PBS with 1% triton x-100 with 1 M NaCl and once with PBS. Between the washes, the beads were collected at the bottom of the tube by 1 min centrifugation at 2000 rpm. Beads were resuspended in a 2x loading buffer, containing 2.5 % β-mercaptoethanol and all samples were denatured at 95 °C for 10 min.

### Immunoblot

Equal amounts of samples were loaded onto NuPAGE® 4–12% gradient Bris-Tris gels (NP0322BOX, Invitrogen) and resolved at 130 V. Proteins were transferred onto a methanol-activated PVDF membrane at 100 V for 75 min in a transfer buffer containing 25 mM tris, 192 mM glycine, and 10% methanol. After transfer, membranes were blocked in 10% milk in TBST (tris-buffered saline with Tween: 150 mM NaCl, 10 mM tris pH 7.5, 0.1% Tween-20) for 1 h at room temperature. They were then washed in TBST and incubated overnight in primary antibody diluted in 3% BSA in TBST. After the incubation, the membrane was washed three times in TBST for 5 min and incubated with fluorophore-conjugated secondary antibodies diluted 1:20000 in 5% milk for 1 h

at room temperature. Prior to imaging on LI-COR Odyssey CLx system, membranes were washed generously in TBST buffer. Quantification of band intensities was performed in Image Studio Lite software and GraphPad Prism 8 was used for the statistical analysis. Source data are provided as a Source Data file.

### Microscopy sample preparation

Cells, grown on 13 mm glass coverslips, were fixed with 4% PFA (Invitrogen, 28906) in PBS for 15 min at room temperature. To remove fixative, cells were washed three times before permeabilisation in 0.1% (v/v) Triton X-100 in PBS or, in case of LAMP1 co-staining, 0.1% (w/v) saponin. Coverslips were then washed again three times with PBS before 20 min incubation with 1% (w/v) BSA in PBS. Coverslips were then incubated with primary antibodies diluted in 0.1 % (w/v) BSA in (with added 0.01% (w/v) saponin in case of saponin permeabilisation) for 1 h at room temperature. After primary antibody detection, coverslips were washed generously in PBS, before secondary detection with Alexa Fluor-conjugated secondary antibodies and DAPI for 1 h at room temperature. The coverslips were finally washed three times with PBS and once in destilled water before mounting onto glass microscopy slides using Fluoromount-G (Invitrogen, 004958-02) and kept refrigerated before analysis on the microscope.

### Confocal microscopy and image analysis

Fixed cells were imaged on Leica SP8 multi-laser point scanning confocal microscope, using a 63x NA1.4 UV oil-immersion lens. Endosomal sub-domains were resolved using Leica 'Lightning' mode for adaptive deconvolution to improve lateral resolution. Leica LAS X software was used for the acquisition of images. Pearson's colocalisation coefficients were determined using Volocity 6.3.1 software (PerkinElmer) with automatic Costes background thresholding. Representative images for colocalisation images were also prepred using the Volocity software. For line fluorescence analysis, high-resolution microscopy images were opened in Fiji software. 30 endosomes from 3 independent experiments per condition were analysed. Briefly, a line was drawn from 'retrieval sub-domain' to the end of 'endosomal core' and fluorescence intensity along line for each channel obtained using 'Plot Profile' function. In R studio software, the fluorescence intensities were normalized to maximum fluorescence intensity of respective channel and endosome. The data was analysed and represented in R studio software using 'ggplot2' package. Representative images for subdomain organisation were prepared in Fiji. Source data are provided as a Source Data file.

### Immunogold labelling and electron microscopy

For Tokuyasu immuno-EM on ultrathin cryosections, cells were fixed with 4% PFA in 0.1 M Phosphate buffer, in 1:1 ratio between fixative and tissue culture medium for 5 min, followed by fixation in fresh 4% PFA for 2 h at room temperature. The fixative was then removed, and cells were rinsed with PBS, followed by 10 min wash in PBS with 0.15% glycine. To detach the cells from a dish, cells were scraped into 1.5 ml of 1% gelatin. After pelleting, 1% gelatin was aspirated and replaced by 12% gelatin at 37 °C. Cells were pelleted again in a swing-out rotor and then the samples were left to solidify in gelatin on ice. Solidified samples were cut into smaller blocks and left in 2.3 M sucrose overnight at 4 °C. Next day, samples were cut into rectangular blocks and stored on pins in liquid nitrogen until sectioning to 85 nm sections at −100 °C on a DiATOME diamond knife in a Leica ultracut cryomicrotome and placed on carbon-coated grids.

For immunogold labelling, grids with sections were washed in PBS for 30 min at 37 °C to remove sucrose and gelatin, rinsed in PBS with 0.15% glycine followed by blocking in PBS with 0.5% fish skin gelatin and 0.1% acetylated BSA. Samples were then incubated with primary antibodies for 30 min and bridging antibodies where

needed, followed by incubation with Protein A conjugated with 10 or 15 nm gold particles. Samples were then fixed in 1% glutaraldehyde in PBS. For the second labelling, the protocol was repeated. Finally, the samples were washed in distilled water, stained for 5 min in 2% uranyloxalate-acetate (pH7), and contrasted with uranylacetate/methyl-cellulose (pH4) for 10 min on ice. The grids were dried by looping out method with a remanium wire loop. Imaging was carried out on Jeol 1011 TEM at 80 kV using Radius software. For each staining and condition, images were aquired from two separate grids. For SNX17-GFP double labelling, a representative micrograph of 9 observed endosomes (2 technical replicates) is shown. For SNX17-VPS35 double labelling, representative micrographs of 15 (parental) and 20 (VPS35L KO) observed endosomes (2 technical replicates) are shown. A more detailed protocol of the immunolabeling procedure was described[50,51].

### AlphaFold2 modelling

All AlphaFold2 models were generated using AlphaFold multimer version 3[27,52] implemented in the ColabFold interface available of the Google Colab platform[28]. The models present in this study include: Retriever (VPS35L, VPS26C and VPS29) in complex with SNX17[400-470] (Model archive: 10.5452/ma-suhwh), SNX17[1-470] in complex with LRP1[4460-4490] (Model archive: 10.5452/ma-k7w97), Retriever in complex with SNX17[1-470] (Model archive: 10.5452/ma-5cyog), D. melanogaster Retriever in complex with SNX17[1-470] (Model archive: 10.5452/ma-w1988) and SNX17[1-470]. 5 independent models were generated for each complex and the quality of the predicted complexes was assessed through examination of iPTM score and the predicted alignment error plot.

### Reporting summary

Further information on research design is available in the Nature Portfolio Reporting Summary linked to this article.

## Data availability

All data and reagents will be made available upon request. Source Data are provided with this paper.

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

## Acknowledgements

We thank the Wolfson Bioimaging Facility at the University of Bristol for their support and Manu Derivery (MRC-LMB) for insightful discussions. We thank René Scriwanek for help with immuno-EM figure preparation. Work in the Cullen laboratory is supported by the Wellcome Trust (104568/Z/14/Z and 220260/Z/20/Z), the Medical Research Council (MR/L007363/1 and MR/P018807/1), the Lister Institute of Preventive Medicine, and the award of a Royal Society Noreen Murray Research Professorship to P.J.C. (RSRP/R1/211004). R.B. is supported by the EndoConnect European Research Training Network (No. 953489) and M.D.H is supported by a Dementia Australia Research Foundation (DARF) project grant. The EM infrastructure in the Klumperman lab used for this work is subsidized by the Roadmap for Large Scale Research Infrastructure (NEMI) of the Netherlands Organisation for Scientific Research (grant number 184.034.014 to J.K.). B.M.C. is supported by an Investigator Grant, Senior Research Fellowship and Project Grant from the National Health and MRC (APP2016410, APP1136021 and APP1156493).

## Author contributions

Cell-based biochemistry and analysis: R.B. Protein purification and recombinant reconstitution: A.P.W., M.D.H. and M.L. AlphaFold2 modelling: R.B., M.D.H. and B.M.C. Immuno-EM: R.B., T.V., N.L. and J.K. Establishment of RPE1-hTERT VPS35L KO cell line: K.K. Manuscript Writing – 1st draft: R.B., B.M.C. and P.J.C; Final Version: all authors. Initial Concept: R.B., M.D.H., K.E.M., B.M.C. and P.J.C. Concept Development: all authors. Funding and Supervision: M.D.H., K.E.M., J.K., B.M.C. and P.J.C.

## Competing interests

The authors declare no competing interests
