## [Peer Review File · Nature Communications]

Mechanism and regulation of cargo entry into the Commander endosomal recycling pathwayREVIEWER COMMENTS

Reviewer #1 (Remarks to the Author):

Over 100 proteins have been previously identified that are regulated by SNX17 and whose retrieval to the plasma membrane relies upon the retriever complex and overall the commander multiprotein complex. Whilst the mechanisms of retriever-cargo interactions have largely been worked out, several significant questions remain to be elucidated, including the mode by which SNX17 binds to the retriever complex, and how SNX17 binding to cargo is regulated (by autoinhibition of SNX17 and competition with NxxY/F motif-containing cargo). In this study, the authors use AlphaFold modelling coupled with a combination of in vitro binding assays, quantitative co-immunoprecipitations and IF + immunoblot analysis of retriever-commander pathway cargo receptors to analyse the mode of SNX17 binding to a retriever interface between VPS35L and VPS26C, as well as provide strong support for an autoinhibitory model in which the SNX17 C-terminal NFAF residues mimic the cargo interaction motif and thus interact with the F3 module of the FERM domain.

Overall, the much of the experimental evidence is convincing and supportive of the conclusions drawn by the authors. Accordingly, the authors make key mechanistic advances in our understanding of events in the sorting of receptors in the endosomal membrane and their inclusion into an important receptor retrieval pathway. There are, however, several points that if addressed would further strengthen the conclusions.

- 1) On Line 147, the authors note that SNX17 E468A and D469A had only a modest impact on retriever association. This statement needs clarifying/editing, as the D469A variant clearly displays a loss of binding to VPS35L and VPS26C (especially, but also VPS29) that is almost complete and more similar to L470G and D467A than it is to the more modestly affected E468A.
- 2) Whilst I agree that Fig. 3 supports a critical requirement for SNX17-retriever binding for efficient receptor recycling/retrieval to the PM, this could be strengthened by dynamic recycling assays. Yes, there are decreased levels of SNX17-driven receptors such as LRP1 and Itgbeta1 at the PM in the absence of VPS35L or in attempted rescue with the R248A mutant. However, the blots also seem to show decreased total (input) levels for LRP1. One might argue that decreased surface receptors lead to overall decreases seen in the input, but since these are chronic knockout cells, it is similarly possible that there is less LRP1 expressed in the KO cells, and thus less is expressed on the PM, regardless of retrieval.
- 3) Please correct typo for “Retriever” in the title of Fig. Legend 3.
- 4) Please correct typo Line 665 in Materials and Methods from “normalies” to normalised.

5) Line 296: Why is Fig. 4B noted here? The statement is accurate based on Fig. 2D, but is not shown in 4D.

6) In Fig. 4F, it would be important to show the controls for equal concentrations of SNX17 and the delta C mutant on the gels.

7) With light microscopy resolution at a maximum of ~ nm in the xy-axis, one concern is whether the authors can be sure that their deconvolution can resolve subdomains in structure of ~200-300 nm diameter. What is the precision of the deconvolution? Details are lacking in the methods. A control demonstrating the resolving power of this system would bolster confidence in the conclusions, or additional superresolution data might be necessary. In addition, whilst there are marker bars, there is no indication as to what size they represent either in the figure (5) or legends.

8) The model in Fig. 5C is sub-optimal. Depiction of the mechanism for SNX17 lipid sensing and and autoinhibition release/cargo binding cargo could be improved and the model would be easier to follow if larger and if the legends could be placed so as to identify the complexes in the figure, rather than below the figure.

Reviewer #2 (Remarks to the Author):

The paper by Butkovic and colleagues examines the molecular mechanism by which cargo enters the Retriever-Commander endomembrane recycling pathway. Using a combination of structural modeling, biochemical studies with Retriever subunit and/or SNX17 variants, and cell imaging, they define key domains important for SNX17/Retriever association and a novel autoinhibitory mechanism where the carboxy-terminal tail of SNX17 occupies the cargo binding domain, thus promoting Retriever association following cargo recognition and binding. This finding, in particular, is timely and important. The Retriever-Commander recycling pathway was first formally discovered in 2017 by this same group, and key aspects of its function remain poorly understood. The current findings extend our understanding of the interaction between the cargo adapter – SNX17 – and the Retriever complex and will thus be of broad interest to the field. The findings create a clear and testable model for the functional association of SNX17 with the Retriever-Commander pathway – cargo binding relieves autoinhibition of SNX17 binding to the VPS35L/VPS26C interface, thus favoring SNX17-Retriever association once relevant cargos have been recognized and recruited by SNX17. The authors provide reasonable evidence in support of their model and future studies will have the capability to incisively test it further.

While the paper has the level of conceptual interest and significance needed for this journal, several issues should be addressed prior to publication.

MAJOR CONCERNS

1) In data supporting the autoinhibition model (primarily Fig4) is based exclusively on a single cargo – LRP1. It would be reassuring to see the same principles emerge with other SNX17 cargos such as itgβ1 and itgα5 for experiments in Fig4B and 4F.

2) Figure 5 shows data with immunofluorescence confocal microscopy that addresses the question of how SNX17-Retrieve and SNX17-cargo interactions affect localization to specific endosome subdomains. The authors conclude that, unlike components of the Retriever-Commander assembly which occupy distinct foci on endosomes, SNX17 is distributed more diffusely across the endosomal surface. This is indeed an important issue, but the data are not particularly compelling given the resolution limits of standard confocal microscopy. The authors should consider super-resolution or ultra-structural approaches so that endosomal sub-domains can be more clearly resolved. This would greatly strengthen the conclusions the authors make.

SPECIFIC COMMENTS

1) The histogram below the blot in Fig2A appears to be mislabeled as the blot shows weaker pulldown with the R293E relative to L35D mutant, while the histogram (as labeled) shows the opposite.

2) The images shown in Fig3A/3B and Fig 4H are low resolution and difficult to resolve details. It is really hard to see differences in endosomal localization among the conditions. Higher quality images should be included.

3) The blot shown in Fig3C is supposed to show surface levels of specific proteins on the right (via surface biotinylation and pulldown) compared to input on the left. But the soluble GFP condition (on the right) after surface biotinylation and pulldown shows a prominent band. Faint bands for GFP-VPS35L fusions and actin are also apparent. Some explanation for why these presumably soluble proteins are being detected in a fraction that is supposed to be exclusively cell surface proteins is needed.

4) The quantitative data in Fig3C expresses the relative surface expression of Retriever cargos in VPS35L KO RPE cells re-expressing either GFP (negative control), GFP-VPS35L (WT), or GFP-R248AVPS35L. It is unclear why the authors chose to normalize the data to N-cadherin (as opposed to WT VPS35L). Some rationale for this should be provided.

5) In the blots shown in Fig4C, the CCDC22 and CCDC93 bands show prominent migration differences among the various SNX17 mutants. This feature of the data is not commented on by the authors. Is there an explanation for these differences?

6) The data in Fig4F shows GST pulldown experiments with WT SNX17 and a SNX17 c-terminal tail deletion mutant. The exact residues deleted are not specified in the text and should be. In addition, it is not clear how many times this experiment was performed. Fig4F shows a single blot without quantification.

7) Some of the experiments include very few replicates (e.g., Fig4G: n=2), yet parametric statistics are performed on the data sets. The meaningfulness of statistics on such a small sample size is questionable.

Over 100 proteins have been previously identified that are regulated by SNX17 and whose retrieval to the plasma membrane relies upon the retriever complex and overall the commander multiprotein complex. Whilst the mechanisms of retriever-cargo interactions have largely been worked out, several significant questions remain to be elucidated, including the mode by which SNX17 binds to the retriever complex, and how SNX17 binding to cargo is regulated (by autoinhibition of SNX17 and competition with NxxY/F motif-containing cargo). In this study, the authors use AlphaFold modelling coupled with a combination of *in vitro* binding assays, quantitative co-immunoprecipitations and IF + immunoblot analysis of retriever-commander pathway cargo receptors to analyse the mode of SNX17 binding to a retriever interface between VPS35L and VPS26C, as well as provide strong support for an autoinhibitory model in which the SNX17 C-terminal NFAF residues mimic the cargo interaction motif and thus interact with the F3 module of the FERM domain.

Overall, the much of the experimental evidence is convincing and supportive of the conclusions drawn by the authors. Accordingly, the authors make key mechanistic advances in our understanding of events in the sorting of receptors in the endosomal membrane and their inclusion into an important receptor retrieval pathway. There are, however, several points that if addressed would further strengthen the conclusions.

We greatly appreciate these very positive and very supportive comments.

1) On Line 147, the authors note that SNX17 E468A and D469A had only a modest impact on retriever association. This statement needs clarifying/editing, as the D469A variant clearly displays a loss of binding to VPS35L and VPS26C (especially, but also VPS29) that is almost complete and more similar to L470G and D467A than it is to the more modestly affected E468A.

We have modified the text to now state: “SNX17(D469A) and, more modestly SNX17(E468A) also impacted on the association with Retriever”.

2) Whilst I agree that Fig. 3 supports a critical requirement for SNX17-retriever binding for efficient receptor recycling/retrieval to the PM, this could be strengthened by dynamic recycling assays. Yes, there are decreased levels of SNX17-driven receptors such as LRP1 and Itgbeta1 at the PM in the absence of VPS35L or in attempted rescue with the R248A mutant. However, the blots also seem to show decreased total (input) levels for LRP1. One might argue that decreased surface receptors lead to overall decreases seen in the input, but since these are chronic knockout cells, it is similarly possible that there is less LRP1 expressed in the KO cells, and thus less is expressed on the PM, regardless of retrieval.

We have included new data utilizing balifomycin to establish that the drop in whole-cell levels of LRP1 and Itgbeta1 is through enhanced lysosomal-mediated degradation (new Supplementary Fig. 2D). These data establish mis-sorting from endosomal recycling to lysosomal degradation and are entirely consistent with our published data establishing that perturbation of SNX17 and the Retriever pathway led to this mis-sorting phenotype (Steinberg et al., 2012; McNally et al., 2017 – data independently validated by other labs).

3) Please correct typo for “Retriever” in the title of Fig. Legend 3.
This has been corrected.

4) Please correct typo Line 665 in Materials and Methods from “normalies” to normalized.
This has been corrected.

5) Line 296: Why is Fig. 4B noted here? The statement is accurate based on Fig. 2D, but is not shown in 4D.

In modifying the text to include the explanation and discussion of new data we have deleted this sentence in order to maintain the flow of the text.

6) In Fig. 4F, it would be important to show the controls for equal concentrations of SNX17 and the delta C mutant on the gels.

In the revised manuscript we have removed the data shown in Fig. 4F. In their place we have included a new series of ITC experiments using purified proteins and synthetic peptides that address the exact same point. Importantly, these new data have provided a more robust biochemical quantification of the association of LRP1 and APP cargo peptides to recombinant full-length SNX17 and a SNX17 deletion mutant lacking the carboxy-terminal tail to relieve the intramolecular auto-inhibited conformation (residues 1-390) (SNX17 Δ C) (see new Fig. 5B and 5C).

7) With light microscopy resolution at a maximum of ~ nm in the xy-axis, one concern is whether the authors can be sure that their deconvolution can resolve subdomains in structure of ~200-300 nm diameter. What is the precision of the deconvolution? Details are lacking in the methods. A control demonstrating the resolving power of this system would bolster confidence in the conclusions, or additional superresolution data might be necessary. In addition, whilst there are marker bars, there is no indication as to what size they represent either in the figure (5) or legends.

We thank the reviewer for making this important comment. After reflection and discussion, we have toned down our emphasis regarding the level of resolution achieved by our light microscopy set-up. We have removed all references to “high-resolution”. The scale bars are now referenced in the figure legends.

However, to re-enforce and independently confirm our conclusions we have used immuno-EM to probe the ultrastructural relationship between SNX17 and the retrieval sub-domain (see entirely new Figure 7). This has confirmed that SNX17 is found on the entire limiting membrane of the endosomal vacuole with the Retromer (VPS35 subunit) and Retriever (VPS35L subunit) being enriched specifically at tubular-vesicular retrieval sub-domains.

8) The model in Fig. 5C is sub-optimal. Depiction of the mechanism for SNX17 lipid sensing and autoinhibition release/cargo binding cargo could be improved and the model would be easier to follow if larger and if the legends could be placed so as to identify the complexes in the figure, rather than below the figure.

We have enlarged the model in a new Figure 8 and modified the figure along the lines recommended.

Reviewer #2 (Remarks to the Author):

The paper by Butkovic and colleagues examines the molecular mechanism by which cargo enters the Retriever-Commander endomembrane recycling pathway. Using a combination of structural modeling, biochemical studies with Retriever subunit and/or SNX17 variants, and cell imaging, they define key domains important for SNX17/Retriever association and a novel autoinhibitory mechanism where the carboxy-terminal tail of SNX17 occupies the cargo binding domain, thus promoting Retriever association following cargo recognition and binding. This finding, in particular, is timely and important. The Retriever-Commander recycling pathway was first formally discovered in 2017 by this same group, and key aspects of its function remain poorly understood. The current findings extend our understanding of the interaction between the cargo adapter – SNX17 – and the Retriever complex and will thus be of broad interest to the field. The findings create a clear and testable model for the functional association of SNX17 with the Retriever-Commander pathway – cargo binding relieves autoinhibition of SNX17 binding to the VPS35L/VPS26C interface, thus favoring SNX17-Retriever association once relevant cargos have been recognized and recruited by SNX17. The authors provide reasonable evidence in support of their model and future studies will have the capability to incisively test it further. While the paper has the level of conceptual interest and significance needed for this journal, several issues should be addressed prior to publication.

We appreciate these very positive and equally supportive comments.

MAJOR CONCERNS

1) In data supporting the autoinhibition model (primarily Fig4) is based exclusively on a single cargo – LRP1. It would be reassuring to see the same principles emerge with other SNX17 cargos such as itgβ1 and itgα5 for experiments in Fig4B and 4F.

We have included a new Figure 4C showing that the enhanced cargo binding upon release of the autoinhibition conformation is also quantified for other classical SNX17 cargo, APP and LRP2.

In addition, to further validate that deletion of the carboxy-terminal tail of SNX17 enhances binding to cargo we have performed ITC experiments (new Figure 5B and 5C). This has confirmed that full-length SNX17 has an approx. 2-fold weaker binding to both LRP1 and APP φxNPxY-containing peptides when compared with the carboxy-terminal tail SNX17 deletion mutant. These new biochemical data provide additional quantitative results supportive of the proposed autoinhibition model.

2) Figure 5 shows data with immunofluorescence confocal microscopy that addresses the question of how SNX17-Retriever and SNX17-cargo interactions affect localization to specific endosome subdomains. The authors conclude that, unlike components of the

Retriever-Commander assembly which occupy distinct foci on endosomes, SNX17 is distributed more diffusely across the endosomal surface. This is indeed an important issue, but the data are not particularly compelling given the resolution limits of standard confocal microscopy. The authors should consider super-resolution or ultra-structural approaches so that endosomal sub-domains can be more clearly resolved. This would greatly strengthen the conclusions the authors make.

In an entirely new Figure 7, we have used immuno-EM to probe the ultrastructural relationship between SNX17 and the retrieval sub-domain. This has confirmed that SNX17 is found on the entire limiting membrane of the endosomal vacuole with the Retromer (VPS35 subunit) and Retriever (VPS35L subunit) being enriched specifically at the tubular-vesicular retrieval sub-domain.

SPECIFIC COMMENTS

1) The histogram below the blot in Fig2A appears to be mislabeled as the blot shows weaker pulldown with the R293E relative to L35D mutant, while the histogram (as labeled) shows the opposite.

Apologies for this mistake which has now been corrected.

2) The images shown in Fig3A/3B and Fig 4H are low resolution and difficult to resolve details. It is really hard to see differences in endosomal localization among the conditions. Higher quality images should be included.

We have replaced these data with the requested high-quality images.

3) The blot shown in Fig3C is supposed to show surface levels of specific proteins on the right (via surface biotinylation and pulldown) compared to input on the left. But the soluble GFP condition (on the right) after surface biotinylation and pulldown shows a prominent band. Faint bands for GFP-VPS35L fusions and actin are also apparent. Some explanation for why these presumably soluble proteins are being detected in a fraction that is supposed to be exclusively cell surface proteins is needed.

We consider this reflects that within the cell-impermeable biotinylation reagents there is a very small percentage that has a level of membrane permeability – we do see batch-to-batch variation in cytosolic labelling from our commercially bought reagent. Hence, we always include the shown controls in every experiment.

4) The quantitative data in Fig3C expresses the relative surface expression of Retriever cargos in VPS35L KO RPE cells re-expressing either GFP (negative control), GFP-VPS35L (WT), or GFP-R248AVPS35L. It is unclear why the authors chose to normalize the data to N-cadherin (as opposed to WT VPS35L). Some rationale for this should be provided.

The level of cell surface N-cadherin does not significantly change within these experiments. As a transmembrane protein control, we therefore normalize to this integral protein.

5) In the blots shown in Fig4C, the CCDC22 and CCDC93 bands show prominent migration differences among the various SNX17 mutants. This feature of the data is not commented on by the authors. Is there an explanation for these differences?

We have no specific explanation for these migration differences. If pushed, we would potential speculate that this may reflect the coincidence in the similar sizes of GFP-SNX17 and endogenous CCDC22 and CCDC93, all approximately 75 kD. As GFP-SNX17 is highly expressed in HEK293T, we believe it may interfere with resolving CCDC22 and CCDC93 proteins on the gel, with mutants W321A and V380D expressing at slightly lower levels than wild-type and M384E (as shown in Figure 4D). This may lead to the observed differences in the migration of CCDC22 and CCDC93.

6) The data in Fig4F shows GST pulldown experiments with WT SNX17 and a SNX17 c-terminal tail deletion mutant. The exact residues deleted are not specified in the text and should be. In addition, it is not clear how many times this experiment was performed. Fig4F shows a single blot without quantification.

In the revised manuscript we have removed the data shown in Fig. 4F. In their place we have included a new series of ITC experiments using purified proteins and synthetic peptides that address the exact same point. Importantly, these new data have provided a more robust biochemical quantification of the association of LRP1 and APP cargo peptides to recombinant full-length SNX17 and a SNX17 deletion mutant lacking the carboxy-terminal tail to relieve the intramolecular auto-inhibited conformation (residues 1-390) (SNX17 Δ C) (see new Fig. 5B and 5C). Exact residue numbers and the number of experimental repeats are stated in the legend.

7) Some of the experiments include very few replicates (e.g., Fig4G: n=2), yet parametric statistics are performed on the data sets. The meaningfulness of statistics on such a small sample size is questionable.

We have performed an additional experimental replicate for Fig. 4G (now Fig. 5D) to bring to a total of n = 3 replicates. We have reperformed the quantification accordingly.

REVIEWERS' COMMENTS

Reviewer #2 (Remarks to the Author):

The authors have provided a thoughtful response to the criticisms from the initial round of review. The changes they have made to the paper have addressed all my previous concerns. The conclusions of the paper are well supported by the data, and this work significantly advances our understanding of cargo entry into the Retriever-Commander endomembrane recycling pathway. The work is impactful and will be of broad interest. The paper is suitable for publication in its present form.

Michael A. Sutton, PhD.

Professor of Molecular and Integrative Physiology

Michigan Neuroscience Institute

University of Michigan